# Development of smart boulders to monitor mass movements via the Internet of Things: A pilot study in Nepal

Benedetta Dini[1], Georgina L. Bennett[2], Aldina M. A. Franco[1], Michael R. Z. Whitworth[3], Kristen L. Cook[4], Andreas Senn[5], John M. Reynolds[6]

[1] School of Environmental Sciences, University of East Anglia, Norwich Research Park, UK; [2] College of Life and Environmental Sciences, University of Exeter, UK; [3] AECOM, UK; [4] GFZ-Potsdam, Germany; [5] Miromico AG, Zurich, Switzerland; [6] Reynolds International Ltd, UK

## 1    Abstract

Boulder movement can be observed not only in rock fall activity, but also in association with other
landslide types such as rock slides, soil slides in colluvium originated from previous rock slides and
debris flows. Large boulders pose a direct threat to life and key infrastructure, amplifying landslide
and flood hazards, as they move from the slopes to the river network. Despite the hazard they pose,
boulders have not been directly targeted as a mean to detect landslide movement or used in dedicated
early warning systems. We use an innovative monitoring system to observe boulder movement
occurring in different geomorphological settings, before reaching the river system. Our study focuses
on an area in the upper Bhote Koshi catchment northeast of Kathmandu, where the Araniko highway
is subjected to periodic landsliding and floods during the monsoons and was heavily affected by
coseismic landslides during the 2015 Gorkha earthquake. In the area, damage by boulders to
properties, roads and other key infrastructure, such as hydropower plants, is observed every year. We
embedded trackers in 23 boulders spread between a landslide body and two debris flow channels,
before the monsoon season of 2019. The trackers, equipped with accelerometers, can detect small
angular changes in boulders orientation and large forces acting on them. The data can be transmitted
in real time, via a long-range wide area network (LoRaWAN®) gateway to a server. Nine of the tagged
boulders registered patterns in the accelerometer data compatible with downslope movements. Of
these, six lying within the landslide body show small angular changes, indicating a reactivation during
the rainfall period and a movement of the landslide mass. Three boulders, located in a debris flow
channel, show sharp changes in orientation, likely corresponding to larger free movements and
sudden rotations. This study highlights that this innovative, cost-effective technology can be used to
monitor boulders in hazard prone sites, identifying in real time the onset of potentially hazardous
movement, and may thus set the basis for early warning systems, particularly in developing countries,
where expensive hazard mitigation strategies may be unfeasible.

## 1.  Introduction

Landslides that affect and originate from mountainous bedrock hillslopes often contain boulders, large
fragments with diameter > 0.25 m and up to several metres. Boulders may have a significant influence
on the fluvial network in terms of landscape evolution, a topic receiving increased attention in the
recent literature (e.g.  Shobe et al., 2020; Bennett et al., 2016). However, the presence in varying
proportions of large grain sizes within a landslide mass can also significantly influence its destructive
power and affect recovery operations. Large boulders can instantaneously destroy properties,
infrastructure and, critically, they can block lifelines for considerable periods of time, as they are the
most difficult component of a deposit to remove (e.g. Serna and Panzar, 2018). Boulders can lie on
hillslopes for a long time (e.g. Collins and Jibson, 2015), before being remobilised as a consequence of
trigger events, such as intense rainfall and earthquakes, which may lead to hazard cascade chains
involving boulder transport. In time, boulders have the potential to move from hillslopes and to enter
debris flow channels and eventually rivers, posing a hazard along the way. Among the far-reaching
effects of boulder movements, damage to hydropower dams can have significant knock-on effects on
local economies (e.g. Reynolds, 2018a,b,c).
The direct and accurate monitoring of boulder movement, also in relation to environmental variables,
is essential in order to achieve a better understanding of the implications of their presence on
hillslopes in active landscapes, the dynamics of their remobilisation and their eventual entrainment in
river systems. In this context, boulder tracking and real-time monitoring represents an important step
forward towards increased resilience in hazard prone areas and it could be performed in different
geomorphological settings, ranging from landslide bodies, to loose slope deposits, to debris flow
channels and rivers, depending on the specific needs and aims. The ability to produce alerts for either
hazardous boulder movements, or to use the movement of boulders to identify hazardous
reactivations of existing large instabilities, requires the careful choice of monitoring techniques that
work in difficult and different environments, preferably wireless and that can reliably send information
in real time. Whilst various early warning systems have been experimented with and put in place for
landslides and debris flows, no early warning system has been used to detect and monitor large
boulders, thus improving resilience with respect to the additional hazards they pose.
Several techniques exist to monitor landslide movements, used also in the context of real time
extraction of displacements. For example, early warning systems have been based on traditional
techniques such as topographic benchmarks, or extensometers, often in combination with more
advanced techniques such as ground based radar interferometry (GB-InSAR) (e.g. Intrieri et al., 2012;
Loew et al., 2017). Geodetic techniques based on GPS or total stations are also widely used and
documented to remotely monitor surface displacements of active landslides (e.g. Glueer et al., 2019).
On one hand, traditional techniques tend to be cheaper but they only allow the retrieval of point-like
information and they can pose challenges for installation. On the other hand, advanced techniques
such as GB-InSAR allow for more continuous coverage but involve much higher costs related to both
equipment and data processing and cannot easily deliver information in real time, even if recent
research has shown the use of radar techniques to deliver real-time data aimed at rockfall hazard
mitigation (Wahlen et al., 2020). Wireless technologies are desirable, due to unfavourable terrain
conditions in which landslide monitoring is often needed. In this respect, passive radio-frequency
(RFID) techniques have recently been used to monitor landslide displacements, and they have been
shown to be inexpensive and versatile (Le Breton et al., 2019). Although this type of technique has not
yet been used in early warning systems, it is contended that the adaptability of such technology could
be developed in this context. The main advantage is their low cost, their wireless nature and also the
ability of the sensors to work in the presence of adverse environmental factors, that would impair
other techniques such as GPS and total stations (e.g. fog, snow, dense vegetation). However, passive
RFID tags currently allow for a monitoring distance (distance between the tags and the receiving
gateway) of a few tens of meters only, which is disadvantageous when monitoring large unstable
slopes or different geomorphic settings in the same area, at the same time. None of the techniques
mentioned above, however, have been used to monitor boulder movement and most of them would
not be suitable for this purpose (perhaps with the exception of passive RFID), thus they have limited
potential in capturing the amplification of landslide hazard posed by the presence of large boulders.
Monitoring movement of sediments within floods has also received much attention in the literature.
For example, bedload transport can be monitored with environmental seismology, in order to detect
the seismic noise generated by moving particles (Burtin et al., 2011; Tsai et al., 2012). Whilst this is
useful in order to identify flood events, or even debris flows events in nearby tributaries, this is also
unsuitable for individual boulder monitoring. Passive radio sensor technology has been used to
monitor movement of individual grains in rivers (e.g. Bennett & Ryan, 2018; Bradley & Tucker, 2012),
however, this technique only allows the quantification of total transport distances between successive
surveys and no real-time data transmission has yet been achieved in this context. Several studies in
coastal settings have tracked individual boulders with extensive field surveys (e.g. Cox, 2020; Naylor
et al., 2016) giving insights into boulder dynamics. Similar efforts to track boulders in fluvial settings
are underway (e.g. Carr et al., 2018). However, such efforts are very time demanding and are also not
suited for real-time detection of boulder movement.
Recently, the use of IMUs (Inertial Measurement Unit) has been tested for different applications in
the field of geomorphology (e.g. Caviezel et al., 2018 and references therein; Frank et al., 2014; Akeila
et al., 2010). In particular, devices able to capture boulder or pebble accelerations and rotations have
been tested in different set-ups in man-made environments. Gronz et al. (2016) have used devices
equipped with a triaxial accelerometer, a triaxial gyroscope and a magnetometer embedded within
pebbles, to reconstruct the path and movement of individual particles in a laboratory flume, with the
aid of a high-speed camera. Such devices, able to capture accelerations up to 4g at 10 Hz, send data
via an 868 MHz radio gateway from where it is then either forwarded to a wireless router or directly
downloaded to a computer via an Ethernet cable. Induced rockfall field experiments were carried out
in the Swiss Alps by Caviezel et al. (2018) in order to test the applicability of IMUs to accurately
measure boulder accelerations and rotations for the calibration of rockfall models. The devices used
in the latter study have high sampling frequency (1 kHz) and acceleration detection range up to 400 g,
the data is stored on a micro SD card and is then downloaded via cable onto a computer. However,
the lifetime of these sensors is limited by battery life (1 to 56 hours, depending on the settings types),
hence requiring development to monitor, in field set-ups, naturally occurring processes, that occur
rarely and unpredictably.
In this study, we aim at filling a gap in the available literature regarding the monitoring of individual
boulders, in real-time and in different geomorphological settings in the field. In the context of the
possible future development of an early warning system, the priority of this pilot study is heavily
focused on capturing the activation of boulder movement in real-time, rather than on the accuracy
and precision of the measurement itself and resolving the full movement, the last two requiring
further development. We explore how displacements or even subtle orientation changes of boulders
lying within a large, slow moving and potentially deep-seated landslide body can be used to identify
landslide reactivation and evolution of the activity levels of different sectors through time. We
contend that this ability may allow to investigate landslide dynamics, geometries and failure modes in
future developments and with denser networks. Additionally, we explore how rapid boulder
movement within active tributary channels could indicate events such as debris flows, and their
monitoring could help identify in the future the forcing thresholds required for remobilisation of
different grain sizes. As mentioned above, technologies that can work in real time and wireless are
better suited for this purpose. For this reason, in this work, we explore the transfer of a technology
developed in the field of ecology to the monitoring of boulders in slow moving landslides and debris
flows. Wireless devices equipped with a GPS module and an accelerometer originally developed for
animal tracking, are modified and adapted for the purpose of boulder tracking and monitoring. GPS
trackers in combination with accelerometers have been used to tag different animals in order to
extract information on migratory, nesting and feeding behaviours among other things (e.g. Soriano-
Redondo et al., 2020; Panicker et al., 2019; Flack et al., 2018; Kano et al., 2018; Gilbert et al., 2016).
Whilst some trackers store the data internally and transmit it to a server via GSM when a network
becomes available, the trackers used for this study have been developed to allow for a network of
nodes that communicate wireless and in real time through an Internet of Things (IoT) system (e.g.
Panicker et al., 2019) that works with an gateway installed locally. In an IoT system, the nodes of the
network communicate to the gateway over radio frequencies and without the need for human
intervention. The gateway can then be directly connected to a computer or, crucially, it can transmit
the data via GSM network to a server in real time.
Transferring this type of technology to boulder monitoring brings several advantages in comparison
to other monitoring systems. The devices used in this work can be used to monitor several boulders
at the same time and in different geomorphological settings within a large study area, thanks to the
longer range achievable by the system in comparison to, for example, RFID techniques. This means
the potential to monitor different hazards (e.g. landslides, debris flows) and different hazardous sites
in the same area, allowing for a comprehensive, simultaneous overview of hazard development
affecting a community and its infrastructure. This also implies the monitoring of several sites within
reach of only one antenna, making the technology cost-effective, and the potential to monitor areas
well upstream of settlements. Moreover, our long-range wireless devices are low-power and can be
directly activated by movement and have real-time communication. These are key features of our
devices and network, since this potentially enables us to 1) develop an early warning system for
hazardous events that involve the presence of boulders, with movement information delivered in real
time and as movement unfolds, 2) monitor during prolonged period without battery replacement (e.g.
one full monsoon season), 3) unravel landslide evolution and mechanics, provided a dense enough
network over a particular site, thus allowing for better evaluation of possible evolution scenarios, as
movement occurs.
In this study, based in the Upper Bhote Koshi catchment (red square in inset in Fig. 1), Nepal, we
demonstrate the use of long-range wireless devices to detect hazardous boulder movement and
landslide reactivation in real time. We also demonstrate for the first time the use of this technology
in the field of geomorphology, and in a field setup, to monitor the movement of boulders embedded
within a landslide and in two debris flow channels.

## 2.  Study area


*2.1.  Hazards and their interactions in the area of study*
Nepal lies at the heart of the Himalayan arc and it is one of the most disaster-prone countries in the
world. In particular, the extreme topographic gradients, seismicity and monsoonal climate, coupled
with increased population pressure (Whitworth et al., 2020), make Nepal widely and frequently
affected by landslides and various types of floods. In 2015 a large number of coseismic landslides were
triggered as a consequence of the Gorkha earthquake sequence, in particular in association with the
largest M 7.8 Gorkha earthquake (25 April 2015) and M 7.3 Dolakha earthquake (12 May 2015).
Several authors mapped coseismic landslides after the events and, although numbers vary greatly (a
few thousands to a few tens of thousands of landslides mapped in different studies), the impact from
these hazards has been unanimously recognised as very significant (Reynolds, 2018b,c; Roback et al.,
2018; Martha et al., 2017; Kargel et al., 2016). The Bhote Koshi catchment, northeast of Kathmandu
(red square in inset in Fig. 1), was also identified as one of the most affected areas, showing the
greatest density of landslides (Roback et al., 2018; Guo et al., 2017; Tanoli et al., 2017; Kargel et al.,
2016; Collins & Jibson, 2015). The areal distribution of landslides away from the main shock epicentre
appears to have been controlled by a combination of PGA, slope and fault rupture propagation
(Roback et al., 2018; Martha et al., 2017; Regmi et al., 2016). Some authors pointed out that many
coseismic landslides occurred at high elevations (e.g. Tanoli et al., 2017), and it was observed that
after the earthquake, a large number of landslides remained disconnected from the channels, with
significant amounts of material stored on the hillslopes (Cook et al., 2016; Collins & Jibson, 2015),
including boulders that are still visible today on valley flanks. During the 2015 monsoon, new
landslides were triggered along with the expansion of coseismic landslides, but loose material
remained stored on the hillslopes by the end of the monsoon (Cook et al., 2016). The sediments
produced with coseismic landslides are expected to move from the hillslopes and into the fluvial
system over several years after the earthquake (Collins & Jibson, 2015 and references therein).
The Bhote Koshi is also highly prone to glacial lakes outburst floods (GLOFs), with six events reported
since 1935 (Khanal, 2015). Different authors have mapped in recent years glacial lakes within the
Bhote Koshi catchment, the total number ranging between 74 and 122 (Khanal, 2015; Liu, 2020),
making glacial lake density in this catchment four times higher than that of the central Himalaya (Liu,
2020). All available studies are in agreement regarding the recent increase in the total area of glacial
lakes in the region, in relation to increasing temperatures and glacial retreat (Liu 2020), with some
authors suggesting that this increase amounts to 47% and that some lakes doubled in size between
1981 and 2001 (Khanal, 2015). Some of these lakes have the potential to drain catastrophically, with
some authors indicating that this risk may increase in the future, as glacial lakes increase in number
and volume. The floods originated from the outburst of glacial lakes can have short-lived discharges
that are several orders of magnitude higher than background discharges in receiving rivers (Cook et
al., 2018) and can have impacts for many tens of km downstream (Richardson and Reynolds, 2000;
Huber et al., 2020; Liu et al., 2020; Khanal et al., 2015). The latest one in the Bhote Koshi catchment
occurred in July 2016, likely originated from a rain-induced debris flow into Gongbatongshacuo Lake,
a moraine-dammed lake in Tibet (Autonomous Region of China) (Cook et al., 2018; Reynolds, 2018a),
that drained catastrophically impacting infrastructure and properties up to 40 km downstream.
Boulders up to 8 m long, weighing in excess of 150 tonnes, jammed the sluices gates of the Bhote
Koshi Hydropower project, diverting the debris-charged flash flood through and totally destroying the
desilting basin, inducing substantial damage to the site (Reynolds, 2018b). During the remedial works
for the reconstruction of the headworks infrastructure, a boulder with 17 m diameter (approximately
4,500 tonnes) was uncovered adjacent to the upstream wall of the headworks dam. This complex
event has highlighted the need for improved ways of understanding the interactions of cascading
hydro-geomorphic processes and to improve measures aimed at increasing resilience (Reynolds,
2018a,c). The availability of loose material on hillslopes, the monsoonal climate and the GLOFs hazard
in the area, enhance the possibility of material containing large grain sizes to reach the river network
via hillslope movements, and eventually be remobilised by exceptionally large floods. Huber et al.
(2020) highlight that very large boulders (around 10 m in diameter) present today in the Bhote Koshi
river have likely been transported by large GLOFs events, supporting the idea that it is unlikely that
monsoon generated floods may have the energy threshold required to remobilise very large grain
sizes (Cook et al., 2018).
Landslides and debris flows can occur also as a consequence of heavy and persistent rainfall during
the monsoon. Every year the area receives up to 4100 mm of rainfall between June and September
(Tanoli et al., 2017). Active monsoons can trigger or reactivate landslides, an example is the Jure
landslide (roughly 15 km southwest of our study sites) occurred in August 2014 (Acharya et al., 2016).
Moreover, intense monsoon rainfall events can trigger debris flows in low order streams channels
within the region (Roback et al., 2018), this allowing for movement of some smaller boulders (> 0.25
m diameter) and allowing hillslope-channel coupling.
*2.2. Geologic and tectonic setting*
Our study sites lie within the Main Central Thrust (MCT) zone (Rai et al., 2017), where the rocks of the
Higher Himalaya Sequence (HHS) are thrusted over rocks of the Lesser Himalaya Sequence (LHS). The
MCT is one of the main faults that accommodate the subduction of the Indian subcontinent under the
Eurasian Plate. The MCT has been mapped at the top and bottom of the roughly 350 m thick Hadi
Khola Schist that is sandwiched between the Dhad Khola Gneiss above and the Robang Phyllite below
at Tatopani, some 5 km upstream of the study site (DMG, 2005, 2006; Rai, 2011; Reynolds, 2018c).
The study site lies entirely within the Benighat Slate, which comprises predominantly black schist,
phyllite, quartzite and carbonate rocks (DMG, 2005,2006; Rai, 2011). The rocks belonging to the HHS
are composed by crystalline, amphibolite to granulite facies metamorphic rocks, mainly ortho- and
paragneisses, quartzite and schists. The LHS rocks present lower grade metamorphism, increasing
towards the MCT, and are largely comprised of phyllites, schists, metasandstones and quartzites
(Basnet & Panthi, 2019; Martha et al., 2017; Rai et al., 2017; Upreti, 1999; Gansser, 1964).

*2.3. Economic assets in the study area – increased vulnerability*

Our study sites are located along the Araniko Highway, a major route that connects Kathmandu to
Kodari and then links Nepal to China. This main road was significantly affected by earthquake induced
landslides in 2015, but is also subjected to landslides every year during the monsoon season (e.g.
Whitworth et al., 2020). The area is of strategic importance for Nepal due to the high concentration
of hydropower projects, either already in operation or under construction (Khanal et al., 2015).
Moreover, the Araniko Highway is a key trade and transport link (Liu et al., 2020) and one of the two
routes between China and Nepal. Khanal et al. (2015) indicate that International trade and tourism
between Nepal and China have been growing rapidly since the opening of the Araniko Highway and
that this route is economically important, with the records of the Customs Office in Nepal showing a
value of US$ 135.9 million in imports and US$ 4.1 million in exports in 2011/2012, with both
governments benefiting from the revenue.

*2.4. Selected sites*

The study site is located at the northern edge of an inferred deep seated gravitational slope
deformation around 1.5 km wide that stretches from Hindi in the north to just upstream of Chakhu to
the south (Reynolds, 2018c).  A secondary landslide body on the northwest-facing valley flank directly
impinging the settlement of Hindi, and two debris flow channels were chosen as tagging sites (Fig. 1).
The most active debris flow channel of the two marks the northeastern boundary of the landslide,
whilst the other channel, which appears to be less active, is located 360 m to the northeast, directly
upstream of the densest part of the settlement of Hindi. Both channels intersect the Araniko highway
and cross the settlement before merging with the Bhote Koshi. The landslide is a soil slide covering an
area of approximately 0.03 km$^2$. Colluvium material likely deposited from previous landslides is visible
at the headscarp and in the terraces along the southwestern flank, with the presence of large boulders
of diameter > 2 m. Large boulders are also observed scattered over the landslide body. The scarp
suggests a depth of the landslide of at least 2 m, and large, fresh cracks were observed in the crown
area in October 2019, indicating activity during the previous monsoon season.

## 3. Methodology


*3.1. Network setup and components*
Twenty-three long range wireless smart sensors, complying with the LoRaWAN® (Long Range Wide
Area Network) specification, provided with external GPS and LoRa antennae and measuring 23 mm by
13 mm (Fig. 2B), were used as nodes in the system. The sensors are equipped with an accelerometer
configured to sample at 2 Hz, as well as a GPS module. In the absence of movement, the devices are
programmed to record and transmit one single location (GPS data only) per day at a fixed time.  When
movement is detected by the accelerometer, so that tilt or acceleration exceed defined thresholds,
collection of GPS and accelerometer data is activated. Different thresholds can be applied for a
detected angular variation in degrees or for a linear acceleration in $g^{-3}$. The values assigned for this
study can be found in section 3.3. The sensors, which were developed by Movetech Telemetry and
Miromico, transmit the acquired data to a LoRaWAN® gateway on the 868 MHz band wirelessly and
in real time. A Multitech IP67 LoRaWAN® gateway, sends the payloads received from the sensors to a
Loriot LoRaWAN® network server through the local GSM network using an agnostic SIM card (Fig.2A-
D). The packages are then sent from Loriot to the Movetech Telemetry server and are decoded
providing the raw information collected by the nodes.
Each sensor was fitted with one (Fig. 2B) or two Lithium C-cells batteries connected in parallel. Twenty-
three boulders were individually tagged by embedding the sensors in a hole drilled in the rock (Fig.
2C). Each boulder was drilled with a 35 mm core drill, for a length of about 15 cm. The depth of the
hole allowed for the emplacement of the C-cell batteries and the sensor. After placement, each hole
was filled with epoxy resin, sealing the cavity, thus protecting the device from tampering and from the
elements (water and humidity), whilst allowing for unaffected connectivity to the gateway via LoRa.
To ease the drilling process but also to allow the epoxy to stay in the cavity before being completely
cured, the holes were drilled at an almost vertical angle (with respect to the global inertial frame), so
roughly from top down. This allowed for the emplacement of the devices flat against the battery inside
the cavity, with $z$ axis near horizontal (global inertial frame), where x and y are oriented as the two
longest sides of the device. There is some variability around the deviation from global horizontal of
the $z$ axes of all our devices, but in general terms the position of the device would follow such setup.
The orientation of the $z$ axis with respect to the cardinal points was not recorded.
The position of the gateway, located in the opposite side of the valley at a distance of about 700 m
from the furthest sensor, at 1330 m a.s.l. and roughly 60 m above the valley bottom was chosen to be
within reach of the GSM network and have direct line of sight with the sensors (Fig. 1 and 2E). Due to
unreliable mains power supply, a 4-panels solar system was developed for this purpose. The initial set-
up did not allow for continuous power to the gateway and led to instability in the system with frequent
offline times during the 2019 monsoon season. However, the system has been improved and it will
guarantee continuous power to the gateway for successive acquisition seasons. The panels currently
charge two 12 V, 110 AH batteries that then provide continuous power to the gateway through a POE
(power over ethernet) supply. The solar system is composed by parts that can be sourced locally, at
relatively low cost and that can be transported to sites without road access, such as the site chosen in
this study. The nature of the local GSM network, relying on one individual antenna in the area at the
time of this study, has also led to frequent GSM connection failures which prevented the gateway
from communicating with the server. The devices deployed in the 2019 season were programmed to
not store the data, but to send it immediately, causing the data transmitted during gateway offline
time to be lost.
*3.2. Choice of tracked boulders*
The tagging sites were selected with the aim of covering different geomorphological settings whilst
retaining visibility to the gateway. The boulders identified for tagging are spread over three sites, two
debris flow channels and a landslide body (Fig. 1). The boulders cover a range of sizes and geologies,
though the geology in this context is not expected to play a significant role in affecting the connectivity
of the network. The smallest boulders tagged have b-axis of 0.3 m, whilst the largest boulder has a b-
axis of 3.3 m (Appendix 1). The selected boulders are characterised by differences in their position at
their location. Boulder location and embedment influenced the choice of the accelerometer settings
used, as explained in the section below. They can be subdivided into three categories: in channel (IC),
partly embedded (PE) and fully embedded (FE) either within the landslide body or in the channel banks
(Fig. 3 and Appendix 2). Boulders in the channel are expected to move freely in case of a large event,
and to be potentially subjected to collisions. Such events could be debris flows with sufficient intensity
to impart forces high enough to cause the boulders to move downslope within the flow. Fully
embedded boulders are not expected to move independently of the surrounding soil mass, as such,
they can only move as a whole with the material on channel banks or with landslide body if these were
to undergo sliding episodes and reactivation (see example schematics in Fig. 5A, B). For these
boulders, generally only the top part is visible, whilst the bottom is fully surrounded by soil. On the
other hand, partly embedded boulders, found at the headscarp, along the southwestern flank of the
landslide or in the channel banks, can either move as a whole with the surrounding material or become
dislodged and begin to move freely on the surface. The second scenario is related to the little amount
of soil covering the bottom part, particularly in the downslope direction, and this scenario would occur
if the soil were to be eroded during intense rainfall events.
*3.3. Sensors settings*
The sensors were programmed to send a routine message every 24 hours, in which only the GPS
position is sent. In between regular fixes the sensors sleep and do not send any data unless movement
occurs, as explained in the following. As mentioned in section 3.1, the sensors can also acquire and
send data in association with an accelerometer event for which activation thresholds can be set for
impact forces and for angular variations. The sensors can be programmed following two main modes:
1) the accelerometer data is averaged over a window of time (over a number of recordings), we call

this mode "*average*" settings (AVG in Appendix 2) and 2) the absolute value of the maximum

acceleration occurring in a time interval can be recorded, and we call this mode "*maximum*" settings

(MAX in Appendix 2). In the first case, the values of the three axes are normalised to $g$ force (where 1

= 1 $g$) and the measurements essentially represent the static angle of tilt or inclination, thus the

projection of the acceleration of gravity, $g$, on the three axes, ranging between 0 (for an axis oriented

horizontally with respect to the global inertial frame) and ± 1 (for an axis oriented vertically with

respect to the global inertial frame). In the second case, the absolute maximum value can be recorded

and this can exceed 1 $g$ and can be set to be as high as 2, 4, 8 or 16 $g$. The measurement resolution

changes according to the chosen detectable maximum, so that a scale capped at 2 $g$ has a resolution

of 0.016 g, whilst a scale capped at 16 $g$ has a resolution of 0.184 $g$ (Appendix 3).

When considering only an individual axis, the variation between two static accelerometer

measurements would correspond to an angular change as shown in Eq. (1):

$$\gamma = arcsin(m/1000) * 180°/\pi \tag{1}$$

where $\gamma$ is the angular variation on a given axis and $m$ is the difference between normalised successive

accelerometer values recorded on the same axis in g. Eq. (1) describes the relationship between

accelerometer output on a given axis and its tilt: for trigonometry, the projection of the gravity vector

on an axis produces an acceleration that is equal to the sine of the angle between that axis and a plane

perpendicular to gravity. According to Eq. (1), if the scale is capped at 2 g, for $m$ = 0.016 $g$ the

corresponding angular variation is of approximately 0.9° if the axis is vertical (with respect to global

inertial frame), but approximately 5.5° if the axis approaches horizontal. Similarly, if the scale is capped

at 16 g, a value of $m$ = 0.184 $g$ corresponds to an angular variation of about 10° when the axis is near-

vertical, but this increases to as high as approximately 21° when the axis approaches the horizontal

(Appendix 3).

The boulders expected to move as a whole with the soil in which they are embedded, and that are
more likely to experience small and gradual angular variations as the surrounding material gently
slides, were programmed with the *average* settings. We chose to cap accelerometer data for average
settings at 2 g (highest resolution), as high impact forces were not expected, and we assigned
thresholds for activation on accelerometer events of approximately 0.4 *g* and 5° for impact forces and
angular changes respectively. The sensors in the two debris flow channels and some of those only
partly embedded within the landslide were programmed to record high impact forces using the
*maximum* settings (Appendix 2). In this case, the scale was capped at the maximum detectable force
of 16 *g* (lowest resolution) and the impact and angular thresholds were set at approximately 4 *g* and
5° respectively. This angular threshold yielded noisier data with respect to the sensors programmed
with the *average* settings type, because of the direct consequence of a drastic reduction in
measurement resolution in the sensors programmed with the *maximum* settings type (Appendix 3),
for which the scale was capped at 16 *g*. Natural measurement variability and errors associated with
the sensors led to spurious data, given the relatively small angular threshold assigned for the highest
detectable maximum of 16 *g*. In other words, given that the step of accelerometer measurement is as
high as 0.184 g, a spurious angular variation of more than 5° is often detected even when the boulder
is stable, due to intrinsic measurement variability (up to 2 bits). Due to the fact that an angular
threshold lower than the scale resolution was imposed, we observed many extra acquisitions triggered
by small variability in accelerometer measurements around a stable value, rather than by true
movement.
In order to reduce the noise in the data due to these fluctuations, a three-stages smoothing is applied
to the raw data. First, a moving window covering 5 successive data points is used. The median value
of the 5 data points is assigned to all points in the window that lie within ± 0.184 *g* of the data point
immediately before the window. If any of the values lie outside the ± 0.184 *g* threshold, then the raw
data points are left unchanged. In the second stage, peaks of one data point are removed (i.e. one
point above or below two points with the same value), this is because if a high impact force is imparted
to a boulder, the position of the boulder is expected to change. This would mean that a high value
would likely be followed by a change in the static angle of tilt of the three axes. Therefore, it is
unrealistic to have a peak value followed by a value equal to that observed before the peak,
particularly when sampling at 2 Hz. This would imply that a boulder undergoes acceleration in one
direction, moves and comes to a halt in the same orientation as before the movement. In the third
and final stage, another moving window of 5 consecutive data points searches for values that lie within
± 0.184 $g$ threshold with respect to the last point immediately before the window.  The same value of
the last point before the window is assigned if all points are within the threshold. If any of the points
lie outside of the ± 0.184 $g$ threshold, the values are left unchanged.
After smoothing, time series of actual accelerometer values were referred to the same zero only for
visualisation purposes, without further manipulation. The accelerometer *x*, *y, z*, values were
recalculated simply as:

$x_t = x_i - x_1$                                                 (2)

for i > 1, where $x_t$ is the transformed, plotted value and $x_i$ all measurements after the first. This allows
the graphs shown in figures 5 and 6 to be analysed more easily, avoiding the y axis scale to be stretched
between -1000 and 1000 mg.
Finally, schematic visualisations of a sample model boulder were produced, calculating pitch and roll
angles changes from the actual data, to indicate the amount of rotation boulders in the channel
underwent (Fig. 6B, D, F). The boulders in the 3D visualisations are, however, extrapolated from the
context of the channel in which they were at the moment of tagging, because it is not possible to
calculate the yaw angle (i.e. the angular variation around the global vertical). The purpose of the
visualisations is just to give a sense of the change in orientation obtained by the boulders between
successive accelerometer measurements (Fig. 6A, C, E), and not that of offering a full 3D
representation of boulder movement.
The sensors are equipped with a GPS module, which is currently also used to retrieve the date and
time of the data acquisition, whilst the data transmission has another timestamp related to the arrival
of the data string to the server. The accelerometer readout in the current version of the software is
tied to a GPS acquisition, this means that although the accelerometer is activated as soon as
movement is detected, the recording of the acquisition is obtained only when the GPS has successfully
retrieved the position. An acquisition of accelerometer data with no GPS position can be obtained and
transmitted (in which case it would only be associated with a server timestamp indicating time of
arrival at the server), but only after the GPS has attempted to retrieve the position and failed. The
timeout for the GPS search has been set to 120 seconds. This is because due to the local topographic
setting and the high valley flanks, the availability of enough satellites at any given time may be low. A
major drawback during the 2019 acquisition campaign was that during the GPS search time, no
accelerometer acquisition can be recorded and transmitted in the current firmware version of the
devices. This means that if boulder movement unfolds over a few seconds, the likelihood is that the
accelerometer recording will only occur towards the end of the movement or after it has stopped
completely, allowing only the retrieval of snapshots of information of two successive static
acquisitions, within seconds (near real time) of the movement starting. Development has already been
made to the firmware to separate the accelerometer acquisition from the GPS for future acquisition
seasons and increase the velocity of accelerometer response to trigger.
*3.4. Validation data*
A Bushnell NatureView HD camera was installed at the gateway location. The camera was set to
acquire an image every 30 minutes and the field of view included the landslide and the southwestern
debris flow channel to around 35 m below the Araniko Highway. Given the rugged terrain and the line
of sight, the visibility in the area around the southwestern flank of the landslide is limited and the
observation is best for the lower part of the slope. Moreover, the plane of the landslide is at a relatively
low angle with the line of sight of the camera. Image cuts were performed for analysis over the visible
parts of the southern channel and of the landslide (Fig. 1). Pixels visually recognisable in all image
frames were manually selected. These correspond to individual trees or boulders and were identified
in successive frames. This allowed for a rough estimate (with an accuracy of about 0.2 m) of the
displacements of these features in the image plane through the available image sequence.
Moreover, the landslide body and the southwestern channel (Fig. 1) were scanned with a Faro Focus
3D X330 terrestrial laser scanner (TLS) in two successive campaigns in April and in October 2019. Each
site was scanned from two scan locations and the point clouds were aligned by matching stable areas
using the Multistation Adjustment algorithm in Riegl RiSCAN Pro (v. 2.3.1). The data were analysed to
obtain ground displacements during the monsoon season, and processed using the point-to-point
cloud comparison method M3C2 in CloudCompare (Lague et al., 2013). Field camera and TLS data
were used to identify days characterised by sliding of the landslide body, sliding of the channel banks,
boulder movements and areas that underwent significant changes of the ground surface. This data is
used in a qualitative way for comparison with and validation of the accelerometer data obtained with
the wireless devices and, despite the qualitative approach, this data provided a quite detailed
overview of the days in which movement occurred. Two Pe6B 3-component geophones recording at
200 Hz were installed on fluvial terraces below the study site to monitor debris flow activity in the
debris flow channels (Burtin et al., 2009).

## 4. Results

We observed that during the 2019 monsoon season, there were important sliding episodes of the
main landslide body (see section 4.1), which caused small and gradual tilt of the tagged boulders
embedded within it. Moreover, although there is no evidence of large debris flows in either of the
channels tagged (for example in the seismometers records), some boulders within the southern
channel bounding the landslide show data that could indicate rapid movement. Of the 23 boulders
tagged, nine show accelerometer time series that are compatible with downslope movement (yellow
to red symbols in Fig. 4). Of these, six lie within the landslide body and were programmed with the
*average* settings in order to detect small angular changes (Fig. 5). The remaining three were located
within the southern debris flow channel and were programmed with the *maximum* settings, to capture
large (> 1 g) impacts (Fig. 6).
In terms of boulder sizes, boulders that appeared to have moved within the landslide have b-axes
ranging from 0.4 to 2.75 m, whilst those that moved in the southern channel have b-axes comprised
between 0.4 and 0.5 m (Appendix 1), thus covering a much smaller range.
The 4 boulders within the landslide that do not show evidence of movement (white circles in Fig. 4),
were fitted with sensors programmed with the *maximum* settings (Appendix 2), due to the fact that
they are partly embedded in the landslide and had potential to become detached from the landslide
body, and thus given the lower accuracy and coarser scale they could not have detected small, gradual
movements even if they had been subjected to them.

466        *4.1 Slow movements within the landslide body*

The movement recorded by boulders embedded within the landslide body is consistent with slow,
gradual tilting that occurred with the sliding of the landslide mass. Small rotational components of the
displacement vector that can either be related to the whole mass or, most likely, to different sectors
of the landslide, induce small angular variations to the boulders embedded within the soil, at the
surface. Fig. 5 shows the accelerometer data for fully or partly embedded boulders programmed with
the *average* settings. The graphs in Fig. 5C-G show the values recorded by the accelerometers in the
*x, y, z* axes through the observation window. Time is shown on the x axis, from 15 May 2019 to 31 Oct
2019, whilst the y axis indicates the value of the projection of *g* on each accelerometer axis in mg (g$^-$
$^3$). The grey curves are raw data and the yellow, orange and red curves are the data after noise was
removed. The data is actual data recorded by the accelerometers, referred to a common zero for
visualisation purposes, as explained in section 3.3 (hence all raw data curves begin at 0, and the
smoothed curves around zero, due to the smoothing). A sketch of the possible type of movement
related to gentle tilting of the boulder within the soil mass, is shown in panels A and B in Fig. 5 and
does not represent any true movement of any of the tagged boulders. The data shows that all sensors
that detected movement were appropriately charged throughout the season (blue curves in graphs).
The variations of the accelerometer axes values from the initial value range from 10 mg to 200 mg in
the different sensors. For an individual axis, the variation in the values would correspond to an angular
change as shown in Eq. (1). Thus, for $m$ = 10 mg, $\gamma \cong 0.6°$ and $\gamma \cong 8°$ for a near horizontal and near
vertical axis (with respect to the global inertial frame) respectively and for $m$ = 200 mg, $\gamma \cong 12°$ and $\gamma$
$\cong 37°$ in the horizontal and vertical cases. In all boulders the rotation is oblique with respect to all axes
and does not occur around any of them.
The images acquired by the timelapse camera (a video is provided in supplements), indicate that the
landslide moved slowly at the beginning of the rainy season and then accelerated later in the season,
most likely in relation to an increase in the pore water pressure within the soil. This temporal evolution
is also observed in our accelerometer data. Moreover, it is likely that the landslide is divided in sectors
with different activity levels and different response to rainfall through time (e.g. Bonzanigo, 2021). In
particular, Fig. 4 and 5 show that the movements of boulders within the landslide not only differ in
the magnitude of the angular variations recorded, which is an order of magnitude higher for B# A226
and B# 9A41 in comparison to other boulders, but also in the evolution with time. Three boulders (B#
33EB, not shown in Fig. 5, B# F3CE and B# 5B6A, the positions of which are also labelled in Appendix
2) show movements early in the time series, already during May and June. The other three boulders
(B# 96F2, B# A226 and B# 9A41) show a later onset of the movement between late August and mid-
September. The boulders with early movements are located below the main scarp (B# F3CE) and in
the middle part of the landslide (B# 33EB and B# 5B6A), closer to the channel, whilst those that move
later are closer to the southwestern flank of the landslide (B# 9A41 and B# 96F2), thus farther away
from the channel, and in the lower half of the landslide body (B# A226).
Visual interpretation of the images acquired by the field camera (section 3.4) indicates that significant
movements of the landslide body occurred during sliding episodes within the orange hatched area in
Fig. 4. The area in which visible changes occurred is about 5000 m$^2$ and corresponds to the lower
portion of the landslide. Fig. 5H indicates the estimated movement magnitudes in the image plane for
the lower, medium and upper parts of the visible sliding area (indicated by L, M, U in Fig. 4).
Displacements roughly up to 2 m in the image plane are detected in the lower and mid-slope parts of
the moving area (Fig. 5H and 7A) between the end of August and the beginning of September, with
upper parts showing displacements of around 1 m. The movement observed in the accelerometer
data of B# A226 and B# 9A41 (Fig. 5F-G) corresponds to the periods in which higher displacement
magnitudes are inferred from the images. Fig. 4 and Fig. 7B also show that boulders B# 5B6A, B# 33EB
and B# 9A41 are located in areas surrounded by displacements as seen by the TLS data (yellow hatched
areas in Fig. 4). Moreover, two boulders within the upper part of the landslide were not found in the
field campaign carried out in October 2019 (B# 33EB and B# 625C), likely due to fresh accumulation of
material from the scarp. Indeed, TLS scan data show cumulative displacements of up to 1 m over large
areas between April and October 2019 (Fig. 7).
*4.2 Rapid orientation changes of boulders in the southern debris flow channel*
Fig. 6 shows the accelerometer data obtained for boulders located within the southern debris flow
channel or on its banks, between 15 May 2019 and 22 October 2019. The graphs in Fig. 6A, B, C contain
the same accelerometer information as explained in section 4.1. The difference in the scale of the
accelerometer output with respect to Fig. 5 is explained by the different settings. These boulders were
programmed to retrieve accelerations higher than 1 *g* (as opposed to normalised values) and forces
up to 16 g. The raw data (grey curves) show frequent oscillations often within ± 0.184 *g* around a value
(corresponding to one step in the accelerometer scale, or one bit) and occasionally up to ± 0.372 *g*
(two steps in the scale, two bits), associated with measurement variability and the coarse scale used
(see section 3.3).
As an example, in the graph for B# 4C02, we observe a change from the initial orientation of the
accelerometer within the boulder equivalent to 1000 mg in y and around 700 mg in x and z. This is
compatible with a change between the initial orientation (1) and orientation 2, attained by the boulder
by 4 June 2019, as visualised in Fig. 6B. The current settings have not captured how the boulder
transitioned between position 1 and position 2, likely due to the very short time interval during which
the change is expected to have happened. The GPS acquisition is likely to have taken longer than the
movement that triggered the recording and delayed the accelerometer acquisition. This applies to the
other two boulders shown in Fig. 6. We do not observe forces > 1 *g* for any of the sensors programmed
with the *maximum* settings, despite the ability of the sensors to detect up to 16 *g*. This is consistent
with a lack of debris flow activity recorded by cameras or seismometers, the more prolonged activity
of which would have generated sustained boulder movement, beyond the time needed for GPS
acquisition as explained below.
Fig. 6G shows rainfall data (daily and cumulative) from GPM IMERG (Bolvin et al., 2015) in green, while
the orange bars indicate days in which movement (sliding of the banks and/or individual boulder
movement) is observed within the channel in the images acquired by the field camera. Often periods
with movement observations occur after days of moderate to intense and/or persistent rainfall. B#
4C02 shows movement data recorded by the accelerometer as early as beginning of June. Even though
this is early in the monsoon season, this movement falls within a few days of moderate rainfall at the
beginning of June during which movements in the channel are already visible in the camera's images.
Similarly, B# 57B9 and B# FB58 show movement (i.e. changes in orientation) that are very close in time
to periods for which other movements are visible within the channel in the images. Just an example
of the several boulder movements observed in the channel in the camera images, a boulder
movement that occurred roughly 25 m downstream of the tagging area in early June is shown in Fig.
8A-B, where two boulders can be clearly seen to move downslope from the banks towards the middle
of the channel by 2-5 m. Fig. 8C shows the areas on the northeastern channel bank and the channel
bed for which significant changes in the ground surface during the monsoon season are detected with
the TLS data. Here, erosion exceeding 1 m is observed in the northeastern bank and accumulation
exceeding 1 m is observed in parts of the channel bed.
The vertical green bars in the graphs of B# 57B9 and B# FB58 (Fig. 6C and E) show the uncertainty
regarding the timing of the recorded movements. Essentially, each green bar indicates a window of
time during which the movement observed may have occurred. The data of each orientation change
marked by a green bar may have been transmitted at a different time from the acquisition time, as
explained below. An explanation of the different scenarios that are described below is also given in
the flowchart in Fig. 9. The orientation change of B# 4C02, the second event of B# 57B9 and the first
event of B# FB58 are characterised by equal GPS timestamp (time of acquisition) and server timestamp
(time of transmission). This indicates that the data transmission occurred within seconds of the data
acquisition (real time). B# 57B9 shows two changes in orientation between 26 and 30 July 2019. The
sensor experienced a gap in the GPS timestamp between 06:15 UTC on 22 July and 06:21 UTC on 28
July, as the GPS failed to obtain a position during this time. Moreover, during this period the gateway
went temporarily offline. Due to these reasons, it impossible to know whether the movement that
caused the orientation change shown in the data transmitted on 26 July occurred immediately before
transmission or during the window for which the GPS timestamp is not available. The gateway
experienced another offline period between 09:36 UTC on 28 July and 03:51 UTC on 30 of July, by
which time the data shows that an orientation change has occurred. Although the acquisitions have
both GPS and server timestamps and these are the same (i.e. acquisitions sent in real time), the actual
movement may have happened at any time between those two timestamps.
During the period encompassing the two recorded movements (26 – 30 July), the field camera images
indicate overcast, rainy conditions that corresponded with important sliding of the right bank of the
channel, offering supporting evidence for movement within the channel. B# FB58 sent data from 15
August 2019 up to 07:17 UTC on 24 August 2019 regularly (based on the server timestamp) but
without a GPS time stamp. A small gap follows, due to the gateway being offline, from 07:17 UTC on
24 August until 16:00 UTC on 25 August, by when the change of orientation has occurred and the GPS
and server time stamp are the same (data sent in real time). Thus, the second movement of B# FB58
is likely to have occurred between these two times, even if the data acquired after the gateway was
online again has been sent in real time on 25 August. The camera images show that movements on
the right bank of the channel occur between 22 and 24 August. The scan data also shows important
displacements in the channel right bank (Fig. 8C). Moreover, 5 boulders in the channel (or on the bank)
were not found in October 2019 at their original location. Two of these are boulders that appear to
have moved in the smart sensors' data and the other three may have been covered by deposition of
loose material.
No boulder movement was recorded for the northern channel, and field observations in October 2019
revealed no signs of recent activity in the channel, which was completely overgrown with vegetation.
*4.3 GPS module limitation*
The GPS had an overall poor performance across all the sensors during the data acquisition season.
The average success rate of GPS acquisition (the ratio between the number of acquisitions with GPS
time stamp and all acquisitions) for the 23 sensors is around 49%, with two sensors never acquiring a
GPS position throughout the time they have been active. Moreover, the standard deviation of
positions ranges between 4.3 m and 15.8 in the x and 5.5 m and 22.6 m in y after removing outliers.
The GPS data acquired is unrealistic not only for the magnitude of the position differences of the same
boulder, but also because the direction is often inverted in time, which is not compatible with possible
boulder movement. However, the poor performance of the GPS for the purpose of boulder tracking
has only limited impact on the ability to detect movement or orientation changes using the
accelerometer, as outlined in the previous sections.
## 5. Discussion
Our data show that nine out of 23 sensors emplaced in boulders at our tagging sites have transmitted
data compatible with real boulder movement, this indicating the potential of the technology used for
detecting both gradual angular variations and changes in boulder orientation associated with rapid
movements in real or near real time. This result, based on the first deployment of this network, is very
promising for the use of this technology in early warning systems in the future, because it shows that
the onset of movement can be identified in real time, provided that all components of the network
operate correctly.
The movements observed for the boulders scattered on the landslide body and embedded within the
material can be described as small angular variations that occurred gradually during the season. Visual
recognition of such movements in the field or in the camera images and scan data would be unfeasible
for individual boulders because they correspond only to small tilt that is difficult to detect with such
methods. However, there are elements that support the fact that the data acquired by the
accelerometers is real and caused by gradual tilting. The images acquired by the camera show
important sliding of the landslide up to 2 m in August-September (see section 4.1 and Fig. 7A), when
the boulders located around the southwestern flank and in the lower part of the landslide show higher
magnitude of the angular variations with respect to other boulders (Fig. 5F, G). The fact that the onset
of movement observed in six boulders in the landslide is not random but appear to follow a spatial
and temporal pattern also supports the idea of a landslide reactivation that causes smaller movements
around the headscarp and nearer the channel to occur earlier. The headscarp activity may not only be
related to the movement of the entire mass, but also to small collapses of the colluvium material in
the steep exposure. This may have led to small movements already from the onset of the monsoon.
Movements in this area are supported by data obtained with the TLS that indicate that displacements
in the line of sight of up to 1 m occurred at or just below the headscarp during the season (Fig. 7B).
Moreover, two boulders in this area were not found in October 2019, most likely because they have
been covered by collapses of loose material from the headscarp. The area near the northeastern flank
may have experienced an increase in pore pressures due to earlier saturation of the soil here than in
the area at the opposite flank, also related to a more rapid increase of the ground water table nearer
the channel driven by topography. We also observe that the magnitude of movements of boulders
closer to the southwestern flank and in the lower slope is higher than elsewhere; this is well supported
by observations obtained through the field camera.
Four partly embedded boulders in the landslide (Appendix 2) were programmed with the *maximum*
settings and showed no movement (Fig. 4). The reason to choose this setting type for these boulders
is that the nature of their position (PE) may have led to larger and faster downslope movements if
they had become dislodged. Given the lower resolution of the data obtainable from the *maximum*
settings, it is possible that nothing is observed for these boulders even if they moved consistently with
the landslide body and experienced slow and gradual tilting of a few degrees. In other words, it is
possible that such boulders also moved but that the nature of the movements may have been too
subtle to be captured with the settings applied. It is also possible that these boulders found
themselves outside of the active sectors of the landslide, although this seems less likely given the
observations obtained in the field and also from camera images and scan data. Although camera
images, scan data and accelerometer data are characterised by different time resolutions, the
movements observed in both landslide and channel in the images and the amount of erosion and
deposition observed in the scan data indicate that the boulders tagged were likely involved in such
movements, and thus there is increased confidence in the fact that the accelerometer data indeed
indicate real movement of the boulders.
Another element that supports the fact that the recorded accelerometer data is associated with real
boulder movement is related to boulder size.  Appendix 1 shows boulder sizes for boulders with and
without movement in the three different tagging sites. For boulders within the landslide body, a size
control on movement was not anticipated. This is because boulders were expected to move as a whole
with the landslide mass and thus their potential to be transported would be independent from their
size. On the contrary, in the channel, and particularly for boulders lying in the channel bed, a size
control on movement is expected, because the size of boulders that could be mobilised by a flow
depends on the flow intensity (Clarke, 1996). Therefore, a flow with low intensity could not be
expected to mobilise the largest boulders tagged. The observations indicate that boulders that show
movements in the landslide are characterised by a much higher range of b-axes than those in the
channel (Appendix 1).
For boulders programmed with the *maximum* settings, we observed noisier accelerometer data than
for those programmed with the *average* settings. What controls this behaviour is not the fact that the
sensors were programmed to detect the maximum force or the static tilt respectively, but rather the
scale that was chosen and associated with the two settings types combined with the choice of angular
threshold to trigger acquisitions. As mentioned before, 16 *g* and 2 *g* were chosen as values to cap the
scale in the *maximum* and *average* settings respectively.
When a sensor is programmed to be capable of capturing forces impacting a boulder as high as 16 *g*,
the resolution currently available for the accelerometer's reading is of 0.184 *g*. Although this is a
relatively small value with respect to 16 *g*, this corresponds to an angular variation of 10.7°. Moreover,
we observe that measurement variability is often 1 bit, but occasionally 2 bits, the latter corresponding
to 0.372 *g* and an angular variation of 21.8°. As the sensors can be activated on both an angular
threshold or an impact threshold detected on any of the axes, care must be taken when selecting the
angular threshold in relation to the achievable accuracy. An angular threshold of 5° at this resolution
is below the measurement error and can trigger a large amount of spurious data strings. This has the
negative effect of diluting the signal with noise and, crucially, to reduce battery lifetime. The downside
of programming sensors with the settings for high impacts recording is that small angular variations
cannot be detected. Future improvements of the accelerometer accuracy, resulting for example from
the activation of the 9-axes IMU present in the hardware of the devices, could reduce this problem.
Although the GPS module is expected to produce readings with a positional error of less than 2 m in
normal conditions, we observed a significant increase in the standard deviation of the measurements
in northing and easting. This could be caused by three effects: 1) the narrow valley drastically reduces
the visibility time of any passing satellites and thus the chances that a suitable number of satellites
will be available to each sensor for calculating the position; 2) the GPS is activated relatively rarely and
this may reduce accuracy (and thus in time precision) of the obtained positions;  3) the rock in which
the sensors are embedded appears to deteriorate the signal. Experiments carried out at the sites have
shown that even sensors placed outside of a boulder, held in the open air and away from obstacles,
needed several minutes to get a GPS position. Moreover, experiments carried out in the UK, at an
open site, have shown that the same sensors at the same site retrieved a position within a radius of
about 50 m when placed inside a boulder and within a radius of about 2 m when held in the open air.
The acquisition of a GPS position is also what causes the largest battery expenditure in the sensors
and it is therefore detrimental for long-term data acquisition on boulder movement. The high
positional errors and the important battery expenditure make the current GPS module not fit for the
purpose of tracking boulders in rugged terrains.
As mentioned above, it is possible to retrieve data strings from the sensors without a GPS timestamp.
So, even if a GPS position, date and time cannot be acquired, the accelerometer data can be recorded
and transmitted anyway, with the server timestamp. In this sense, the fact that the accelerometer was
tied to the GPS during the 2019 acquisition season, so that the accelerometer data could be recorded
only once the GPS acquisition has been attempted and failed, did not invalidate completely the data
output.
However, there are also important limitations related to this. As the time for the GPS acquisition
attempt was set to 120 seconds, the sensor measures the acceleration already during this time, but it
does not record it nor transmit it until the GPS position is either acquired or fails. In the case of fast
movements, or relatively large impacts caused by the sudden movements of boulders within the flow,
120 seconds (this would often be even more, in case a GPS acquisition is being obtained) may be
enough time for the movement to begin and stop. This may explain why, although the boulders in the
channel were programmed to detect high forces, they never show accelerometer values higher than
1 $g$ (either negative or positive). In essence, these sensors have also only recorded the static tilt and
different orientations acquired by the boulders in time (within seconds of movement occurrence), but
not the actual movement as it unfolded. For instance, the position change of B# 4C02, B# 57B9 (second
event, i.e. event that causes transition from position 2 and 3) and B# FB58 (first event, i.e. event that
causes transition from position 1 and 2) were received in real time. This means that as soon as the
data string indicating a different orientation with respect to the previous data string was acquired, it
was also sent. In this type of situation, the GPS timestamp is the same as the server timestamp, but
there is no recording of the movement as it unfolded. The event of B# 4C02 points to the fact that the
GPS delayed the acquisition of the accelerometer data, because the gateway was online during the
time in which the orientation change must have occurred. Given that there is no evidence of large
debris flows during the 2019 monsoon season, B# 4C02 may just be one example of minor boulder
movement that started and stopped within the 120 seconds time interval. This may be improved in
successive acquisition seasons, since development has been made in order to separate the GPS from
the accelerometer acquisitions. The next batch of devices that will be deployed in the network will
thus be able to capture faster rotation already from the start of the movement.
The picture may be complicated even further by the fact that occasionally the gateway experienced
some offline time, due either to the battery not being recharged properly or to GSM connection loss.
This is the case of B# 57B9 (second event) and B# FB58 (first event), in which we observe that the data
string indicating an orientation change is sent in real time, but follows a gap in the gateway
connectivity. In this case, the movement may have occurred at any point during the offline period of
the gateway, then the first acquisition since the gateway became once again online is sent in real time.
However, a new solar system is now in place and will prevent future power issues during future
acquisition seasons. Finally, the accelerometer sampling acquisitions that could be reached in the 2019
campaign was 2 Hz. While this is acceptable to detect gradual angular variations that occur slowly over
a prolonged period and allowed us to identify periods of acceleration of the rotations, it is too low if
the aim is that of capturing a fast movement in the channel. For this reason, the capability of our
devices has now been increased to record data up to 400 Hz.
*5.1 Advantages and limitations of this technology*
The LoRaWAN® smart active sensors developed in this study for the purpose of identifying boulder
movements has already shed light on its potential advantages and its limitations. The technology used
is independent of weather conditions. The communication between the tags and the gateway is not
hampered by adverse weather conditions and movements were observed during overcast and rainy
days.  This is of course true if the gateway is powered with batteries of sufficient capacity to withstand
days with insufficient sunlight, which may occur during the monsoon season. Although a good visibility
of the sensors from the gateway increases connectivity between the nodes and the gateway, the long-
range nature of the system allows for a network that extends over a relatively large area. In our case,
we were able to obtain data from boulders located at up to 800 m from the gateway, covering an area
of about 0.25 km$^2$, this likely not being the upper limit of the achievable range. This is especially
advantageous for a number of reasons. Different geomorphic features can be monitored with the
same gateway, in our case including a landslide and two debris flow channels. Moreover, in
comparison with other innovative and promising techniques such as passive RFID technology (Le
Breton et al., 2019), which can currently allow for a range of about 60 m, our network offer the
advantage of covering different sectors of the main landslide, in case of large unstable areas, thus not
limiting the observation to restricted sectors, which could offer a more complete picture of the
instability dynamics. Moreover, the long range of our devices can allow to increase the monitoring
area further, thus potentially enabling us to identify movement further upstream in the monitored
channels (provided feasibility in drilling into boulders in active sites), which is essential to provide
enough lead time to secure operations at major infrastructure sites or to alert downstream
populations.
An important characteristic of the devices used in this study as opposed with other techniques is that
they are active and can easily be assigned thresholds (e.g. acceleration or tilt) that can be used in an
early warning system context. Moreover, the devices can be embedded directly inside boulders,
without the need for additional supports that may 1) make the devices more visible/exposed and thus
more subjected to intentional tampering or animal damage, 2) there is no additional movement to be
accounted for (e.g. tilting of supporting poles). The technology is also relatively low cost and has the
potential to become competitive and cost-effective in the future. The most expensive component is
the gateway (around 1000 USD), whilst the devices are around 200 USD each. The ability to retrieve
the tags after battery consumption has already been investigated and will be implemented in
successive acquisition seasons, will allow for a durable, cost-effective network. This may make this
technology more affordable than other more expensive techniques such as GB-InSAR, GPS or total
stations and can allow dense networks.
The main drawback encountered in this study is the poor performance of the GPS module, which made
it impossible to directly evaluate the magnitude of displacements either of the landslide or of
individual boulders. Measurements of displacement are ideally needed to understand landslide
velocity changes in time and space for example in response to climatic forcing (e.g. Handwerger et al.,
2019; Bennett et al., 2016) as well as to identify the acceleration of a landslide towards failure (e.g.
Carlà et al., 2019; Handwerger et al., 2019). Moreover, the GPS acquisition, tied to the recording of
accelerometer data, has hampered in some cases the ability to obtain the full sequence of
accelerations experienced by the boulders. This issue will however be resolved in the next acquisition
season, since further development has allowed us to make the accelerometer independent of GPS
acquisitions. Work is also planned to write the firmware to enable the gyroscope and magnetometer
on the device, which will give more detail of boulder dynamics such as rotations. Finally, the
connectivity of the gateway to the server (during offline periods) has prevented some of the time the
ability to receive the movement signal in real time. This problem has now been resolved, with a more
stable solar system currently powering the gateway, thus future acquisition seasons should benefit of
higher robustness and less connectivity loss.
**6.** Conclusions
We show the application of a smart sensor LoRaWAN® network for the detection of boulder
movements within a landslide and a debris flow channel in the Upper Bhote Koshi catchment
(northeastern Nepal). We tagged 23 boulders ahead of the 2019 monsoon season with devices
equipped with an accelerometer and able to send data in real time to a LoRaWAN® gateway. Of these
23 boulders, nine sent data compatible with movement. Six of these were fully or partly embedded in
a soil slide and are characterised by accelerometer time series that indicate slow, gradual angular
variations. Such angular variations reflect the movement of boulders within the landslide mass. The
reactivation of the landslide is confirmed by both timelapse cameras and TLS data. Also, the
movements show staggered onset, so that the boulders nearer the scarp or the lower boundary, near
the channel, began to move earlier in the season than other boulders. In the channel, only three
boulders show data likely corresponding to sharp, sudden movements and rotations that occurred in
response to intense or persistent rainfall. The sizes of the boulders that moved in the channel are
towards the smallest end of the boulders tagged in the channel, reflecting the fact that no large debris
flows were observed in the channel during the 2019 monsoon season.
Though with some limitations, the technology has proven able to detect boulder movements with this
type of device, for the first time in a field setup as opposed to a laboratory setup. In the optimal
conditions of all the component of the network operating properly, the ability to capture the onset of
movement in real-time is an important premise to the use of this technology in early warning systems
of slope movements that involve the presence of hazardous boulders. This pilot study also hints at the
potential of these devices to further understanding of landslide dynamics, for example the timing of
movement in response to rainfall and the spatial sequencing of movement across a landslide. The
most important challenge that we believe has prevented the recording of the complete movement for
the boulders in the channel is related to the current requirement for a GPS position to be acquired for
the accelerometer data to be recorded and transmitted. Furthermore, the poor GPS performance
currently precludes the measurement of displacements. However, the sensors are already equipped
with a 9-axis IMU comprising an accelerometer, a gyroscope and a magnetometer, that have not been
ready for the field tests in Nepal, that might allow the retrieval of more information on movement,
when combined with field observations and optical images.
Future work will involve the tagging of more boulders at the same sites of the current network to
improve the accelerometer sampling frequency, the now improved the stability of the network
connectivity, more suitable programming settings and the ability to retrieve and reuse the tags.  In the
next batch of devices, we will be able to activate the accelerometer and record movement data
independently of the GPS acquisition. This is expected to significantly speed up data acquisition and
transmission to the server, which will be a step forward in view of using this technology for early
warnings. Moreover, this will also allow us to capture the whole accelerations sequence associated
with fast rotations induced by large impact forces and may enhance the understanding of boulder
movement from the hillslopes into the river network.
Acknowledgments
This work was carried out as part of the BOULDER project, funded by the NERC/SHEAR Catalyst
program (NE/S005951/1). Nick Griffin carried out essential work related to powering the devices and
setting up the solar system. Gareth Flowerdew indefatigably carried out the drilling, essential for
embedding the devices in the boulders. Phil Atkinson has contributed to this work by helping decoding
the raw data and managing SIM card usage of the gateway. Shuva Sharma and Pawan Timsina from
Scott Wilson Nepal (SWN) provided support during the initial phases of the work, network installation
and helped organising dissemination workshops for the project. Bhairab Sitaula's contribution to
logistical and technical aspects of the field campaigns was essential. Bibek Raj Shreshta contributed to
boulder tagging and Joshua Jones helped finding the tagged boulders after the monsoon. Luc Illien
helped placing the seismometers for detection of debris flows for validation of our data. Alan Rae and
Stephen Drewett provided support related to the LoRaWAN® server and the gateway. Stephen
Laycock at UEA shared a code to visualise our accelerometer data and the orientation changes with a
model boulder.

**Author Contributions**
B.D. tested and programmed the sensors, analysed the data and wrote the paper; G.L.B. shaped the
idea, wrote the proposal obtaining funding for this work and contributed to the data analysis; A.M.F.
tested and programmed the sensors and contributed to the data analysis. B.D., G.L.B., A.M.F. and
M.R.Z.W. carried out field work and network installation. C.L.K. installed the seismometers, carried
out the two scans of the area and contributed to the analysis of the scan data. A.S. carried out software
development and participated to field work. J. M. R. contributed to the original idea of the project. All
authors revised and made contributions to the manuscript.

**Competing Interests statement**

The authors declare no competing interests.

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

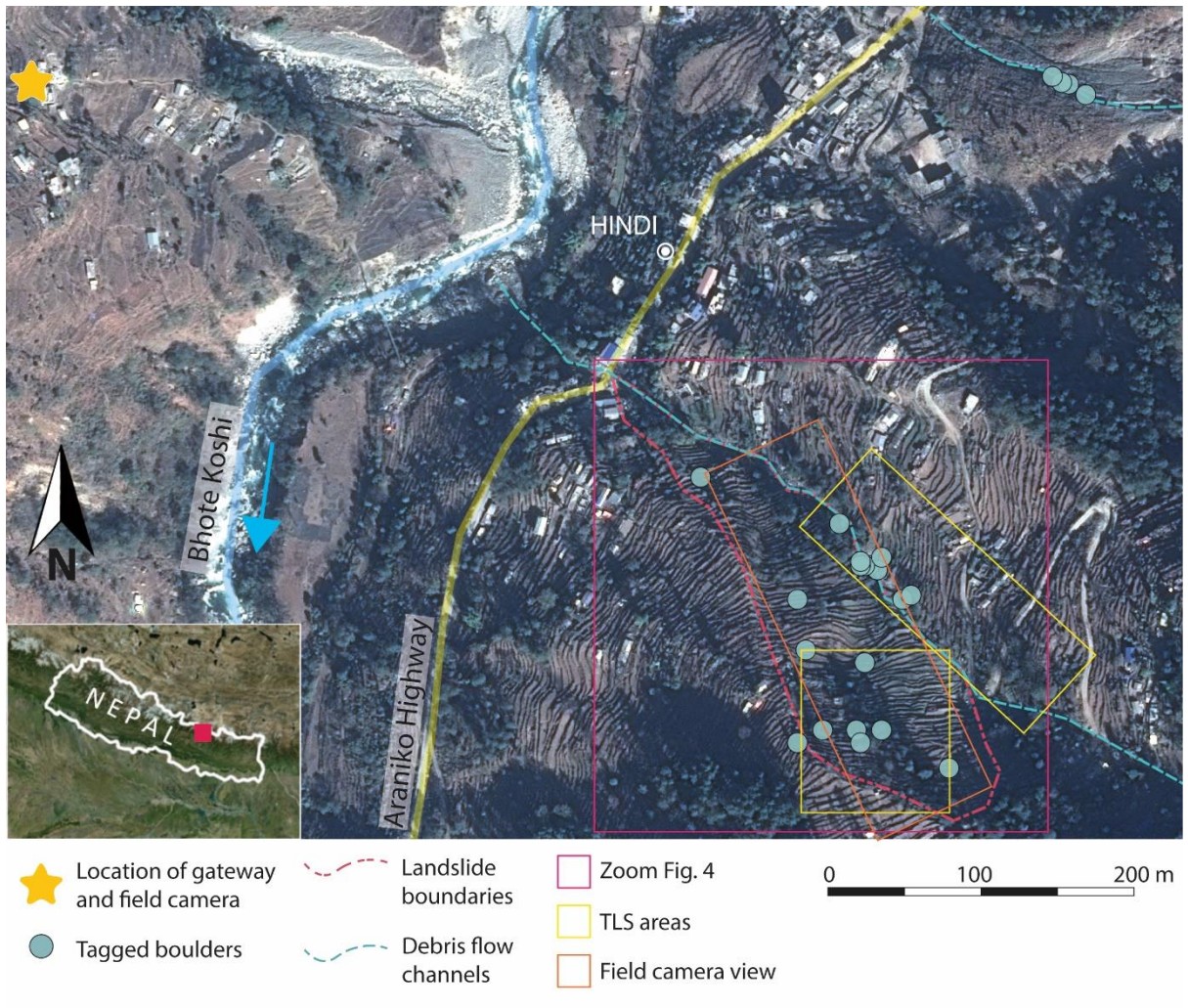

Fig. 1. Overview of study area and network, including three tagged sites (two debris flow channels and a landslide body). Red box, zoom of two tagged sites. Yellow boxes, terrestrial laser scanner areas. Orange box, field view of field camera. Image: Pleiades (CEOS Landslides Pilot).

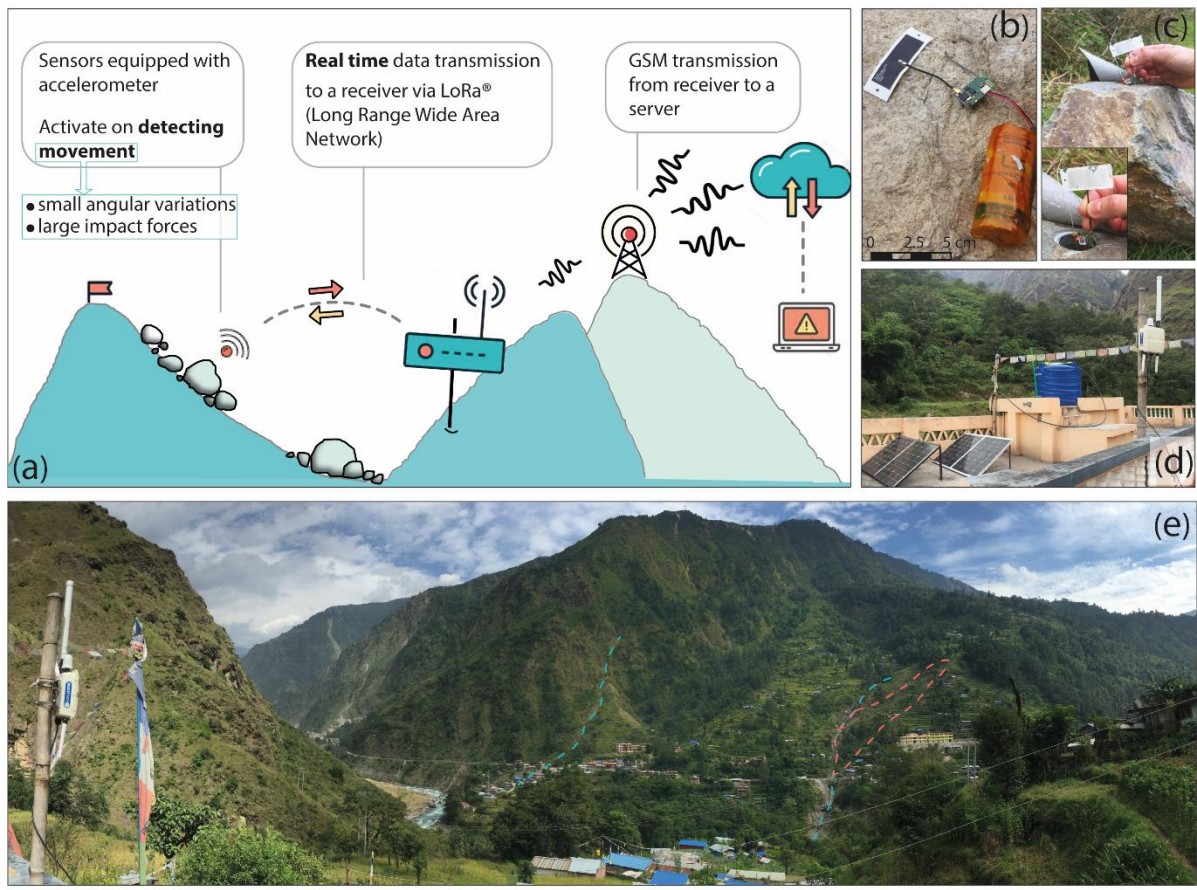

Fig. 2. A) Sketch of the network, its components and communication methods. B-C) Sensor and tagging of a boulder. D) Gateway setup. E) Overview of the tagging sites from the gateway. Gateway visible in the far left of the image. Blue dashed lines mark the debris flow channels and red dashed lines mark the boundaries of the landslide.

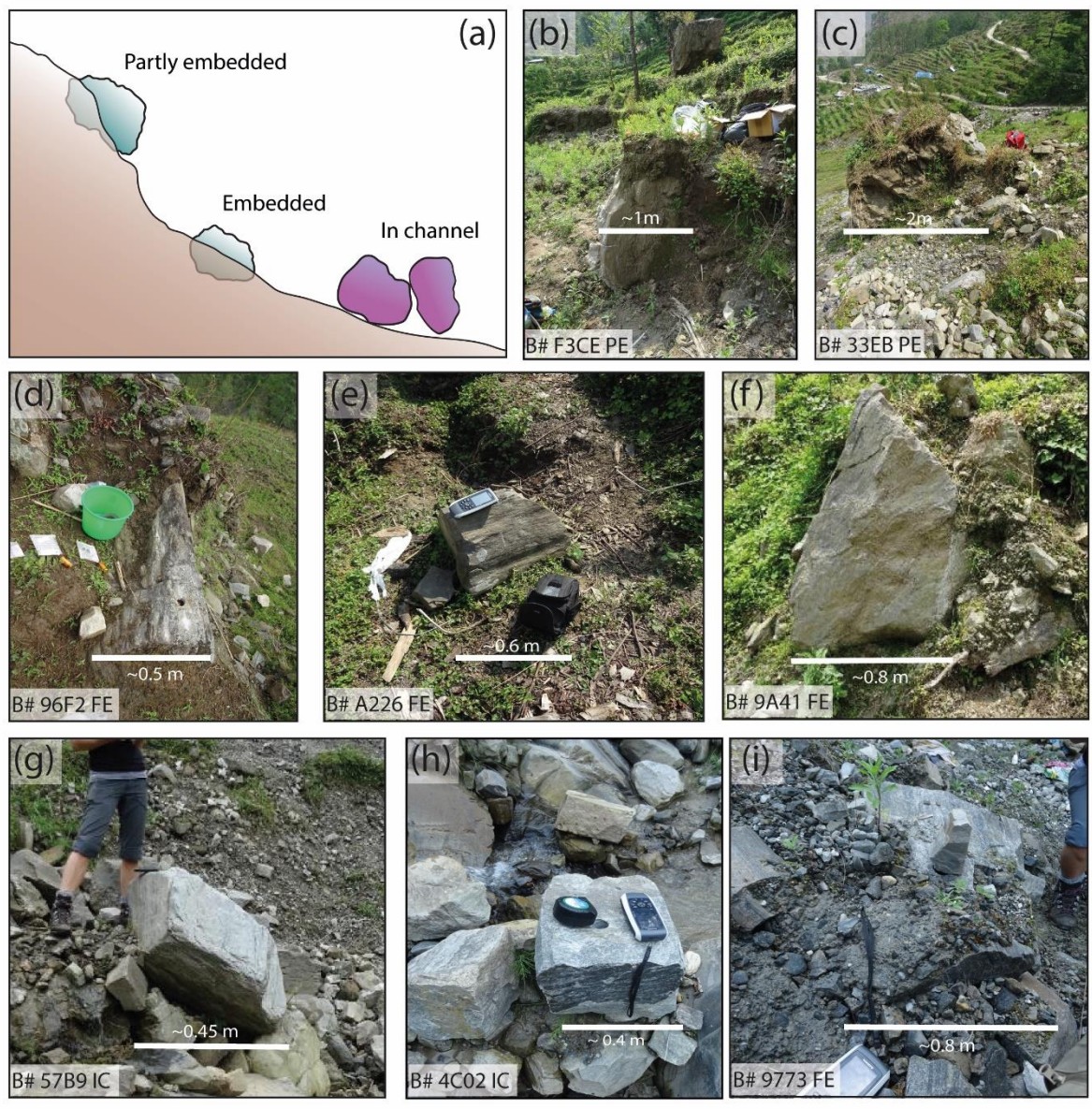

Fig. 3. A) Sketch of boulder position types. B-C) Examples of partly embedded (PE) boulders within the landslide body. D-E-F) Examples of fully embedded (FE) boulders within the landslide body. G-H) Examples of boulders inside the main channel (IC). I) Example of fully embedded (FE) boulder within the channel bank.

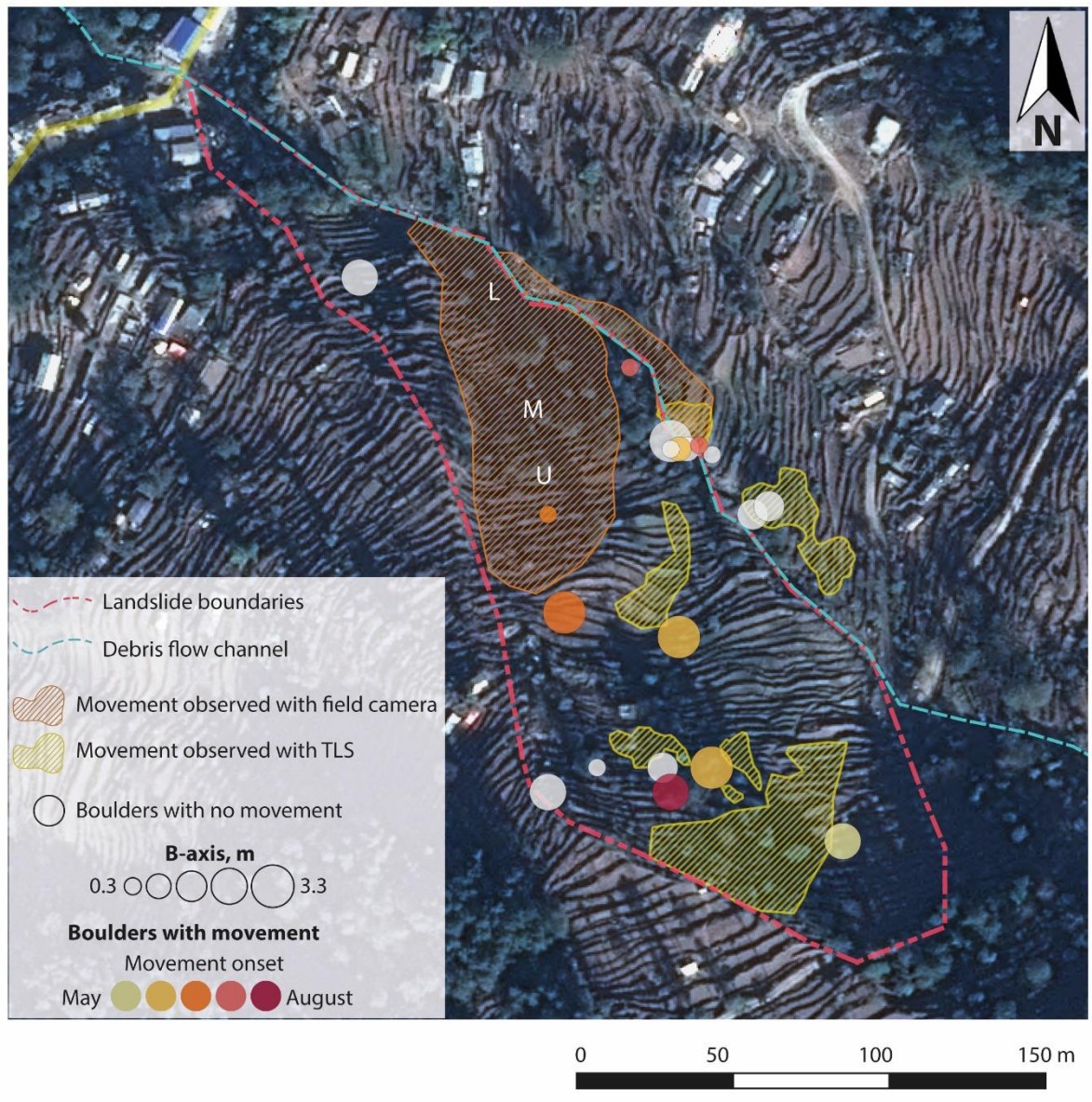

Fig. 4. Zoom of two tagged sites. The sizes are scaled according to the b-axis of the boulders (example of scales given for boulders without movement in legend but applies to all boulders). White squares are boulders that did not move or for which movement was not recorded. Green circles are boulders in the debris flow channel. Yellow to red symbols are boulders within the landslide body. Hatched areas are zones with observed movement through images (L: lower, M: mid-slope, U: upper). and terrestrial laser scanning. Image: Pleiades (CEOS Landslides Pilot).

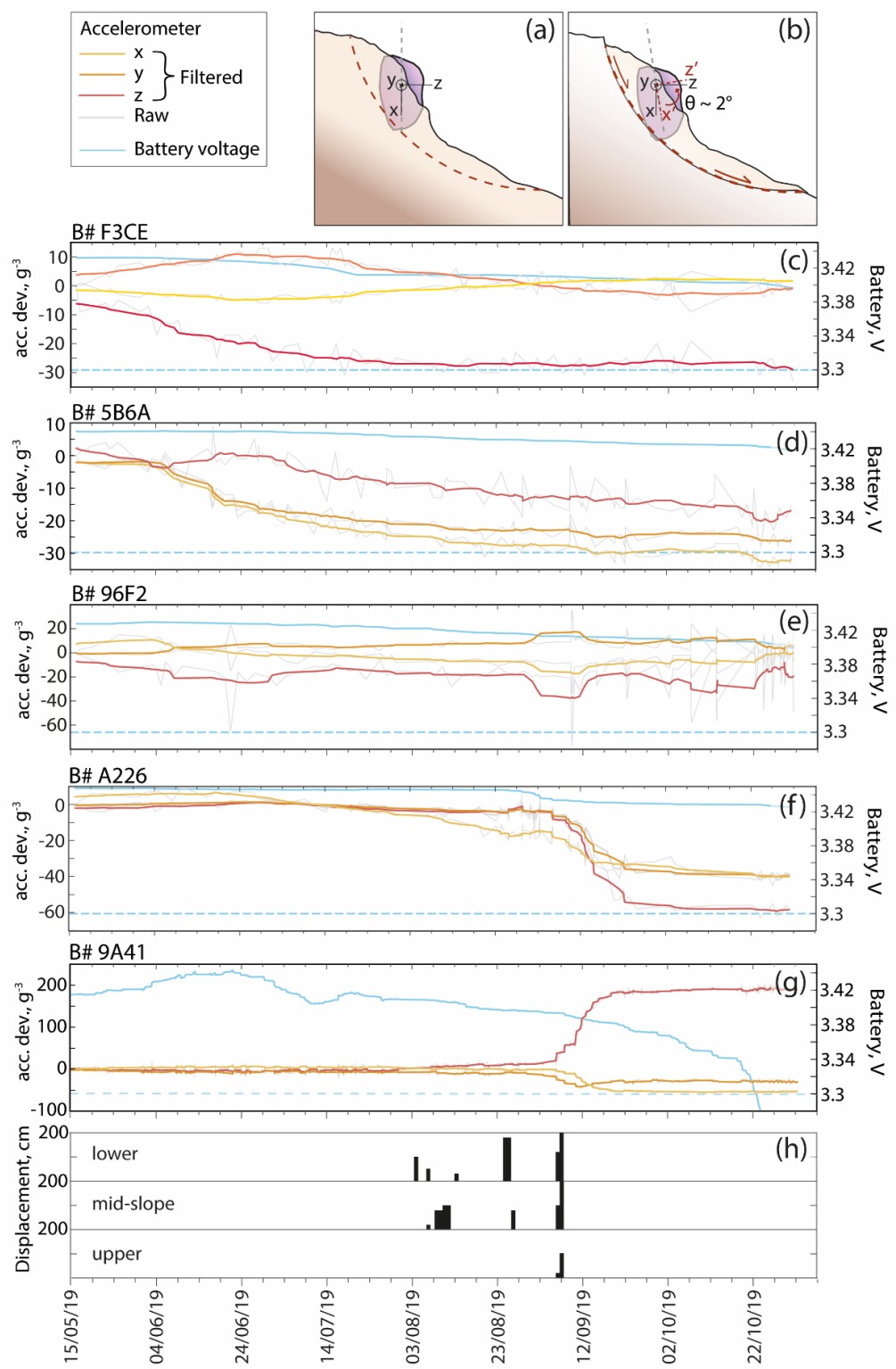

Fig. 5. C-G) Real accelerometer data (raw and smoothed) showing deviation from initial position for each axis for boulders within the landslide body through the monsoon season. A-B) Sketch of possible type of movement experienced by embedded or partly embedded boulders. Note that this is only a schematic to indicate a movement that occurs in accordance with the landslide body and does not necessarily represent real movement of the boulders monitored in this study. H) Estimated displacements of lower, mid-slope and upper parts of the slope obtained through field camera images. The yellow, orange and red curves in the line plots (Fig. 5C-G) represent the smoothed data of the accelerometer *x*, *y*, and *z* axes respectively, the grey curves represent the raw data for each axis. The blue curve shows the battery voltage, and the blue horizontal dashed line represent the 3.3 V threshold below which the battery is discharged and faulty behaviour may be expected.

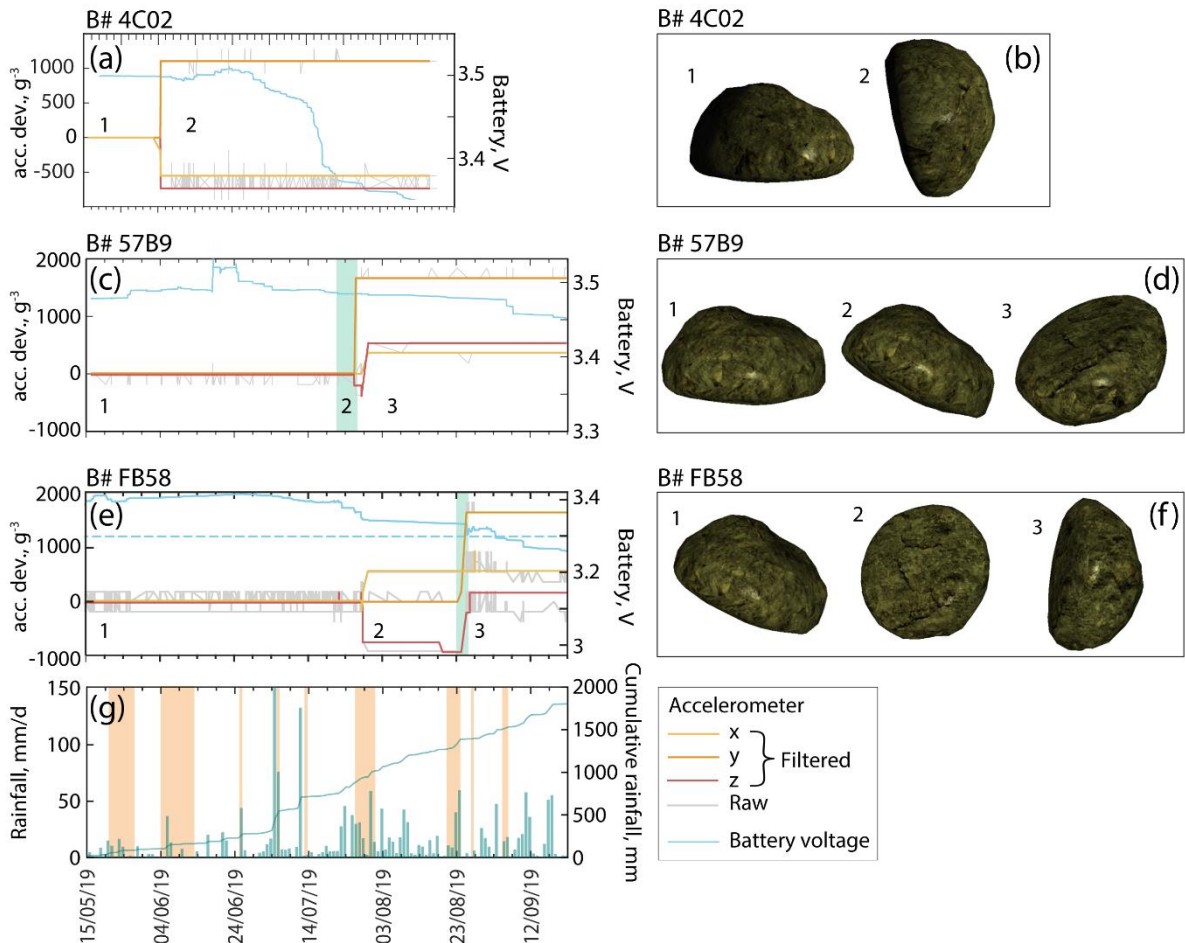

Fig. 6. A,C,E) Real accelerometer data (raw and smoothed) showing deviation from initial position for each axis for boulders in the debris flow channel and its banks through the monsoon season. Light green bars represent uncertainty in the movement timing due to lack of GPS acquisition (i.e. no time recorded) or offline gateway. G) Daily and cumulative rainfall data from GPM. Yellow bars represent days in which movements are observed in the channel and/or on its banks in the field camera images. B,D,F) Model boulder 3D visualisation to represent the change from the initial positions of the boulders and the positions acquired after the recorded movement, only in terms of pitch and roll angles. Note that the boulders are in a space with no coordinates, because the visualisations do not indicate the position of each boulders within the channel, but only the pitch and roll angle changes. Numbers of positions are marked in the accelerometer graphs.

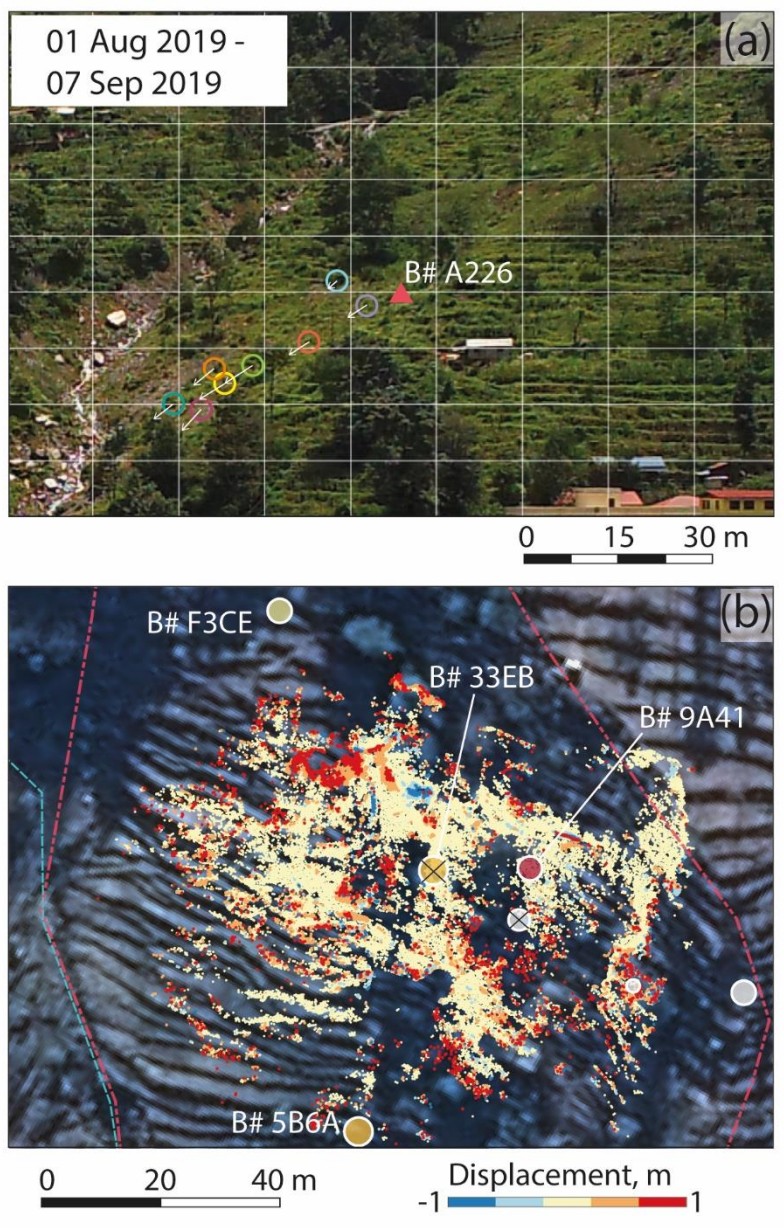

Fig. 7. Examples of movements in the landslide body between A and B. Coloured circles represent visually traceable pixels. Their movement is visible through the superposed grid. Approximate location of B# A226 is shown. C) Scan data for the upper part of the landslide area shows several zones of movement, where red represents accumulation and blue erosion. Black crosses over the boulders represent boulders that were not found after the monsoon season. Image: Pleiades (CEOS Landslides Pilot).

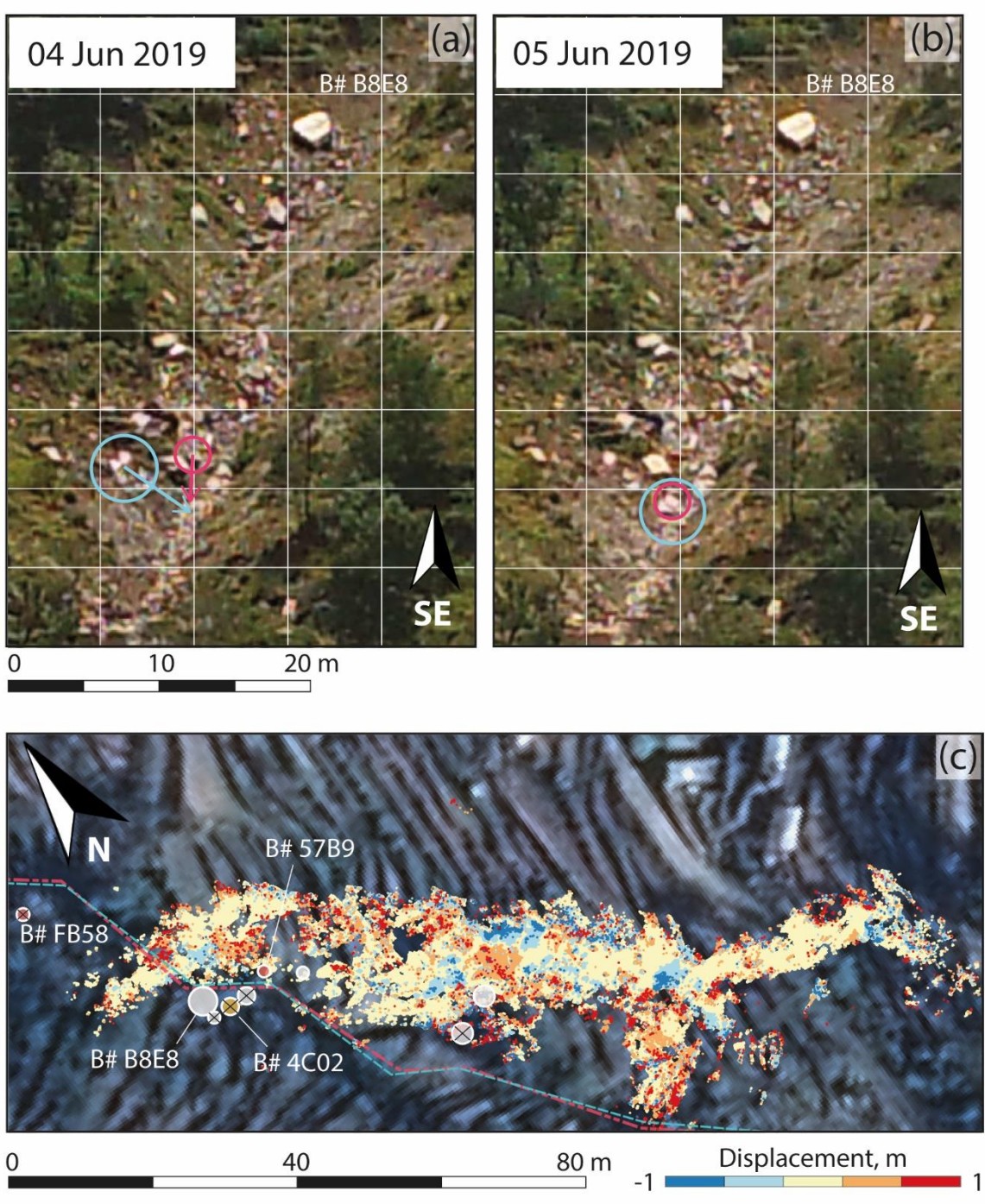

Fig. 8. Example of movements in the debris flow channel between A and B. Example of movements in the channel banks and in the channel between C and D. Coloured circles represent traceable pixels. Coloured boxes represent areas in which large changes are observed. E) Scan data for the channel showing several zones of movement, blue represent collapse of parts of the orographic right bank, red represents accumulation areas. Black crosses over the boulders represent boulders that were not found after the monsoon season. Image: Pleiades (CEOS Landslides Pilot).

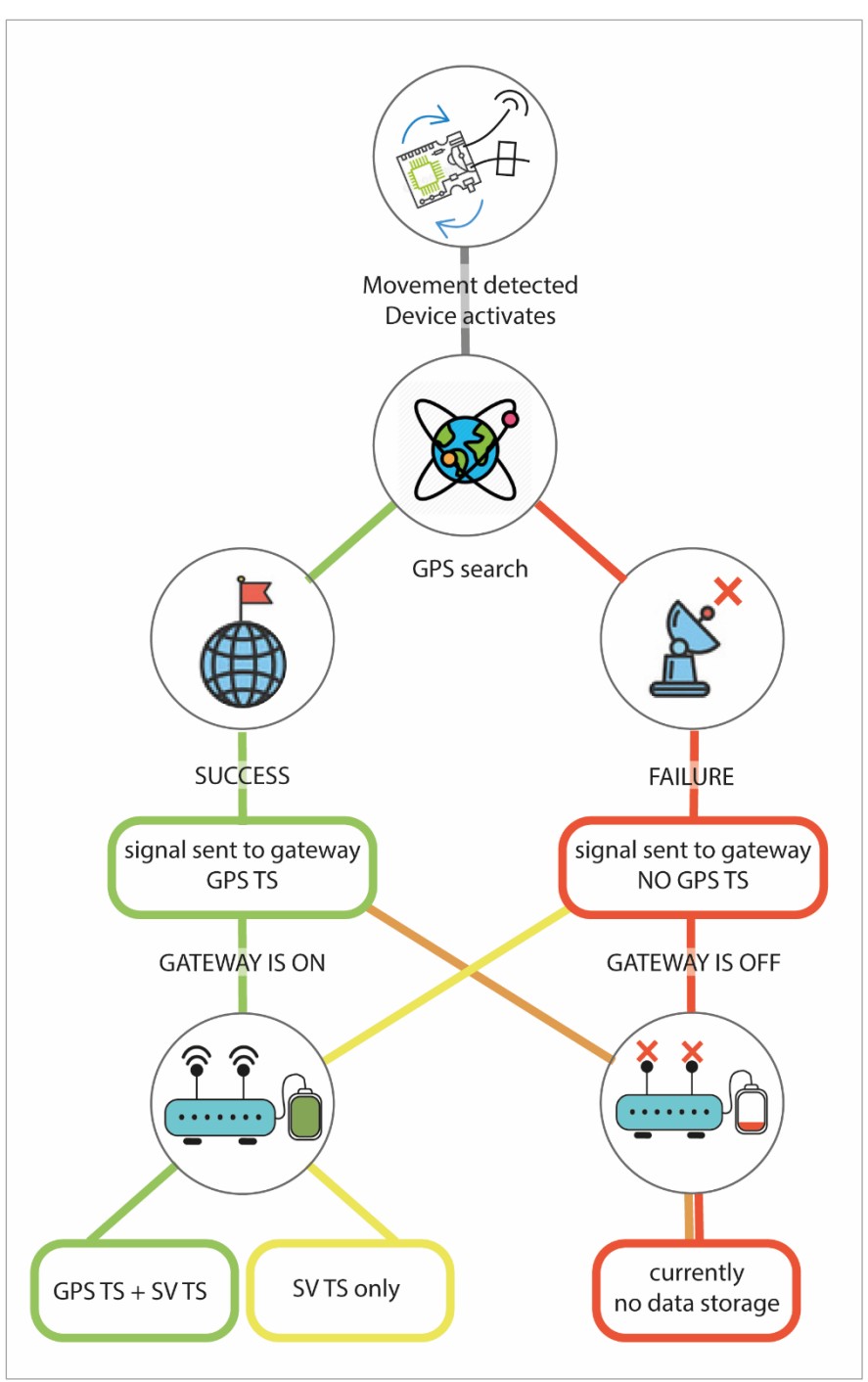

Fig. 9. Flowchart illustrating the presence of GPS timestamp (GPS TS) and server timestamp (SV TS) and the different scenario of GPS acquisition and data transmission.

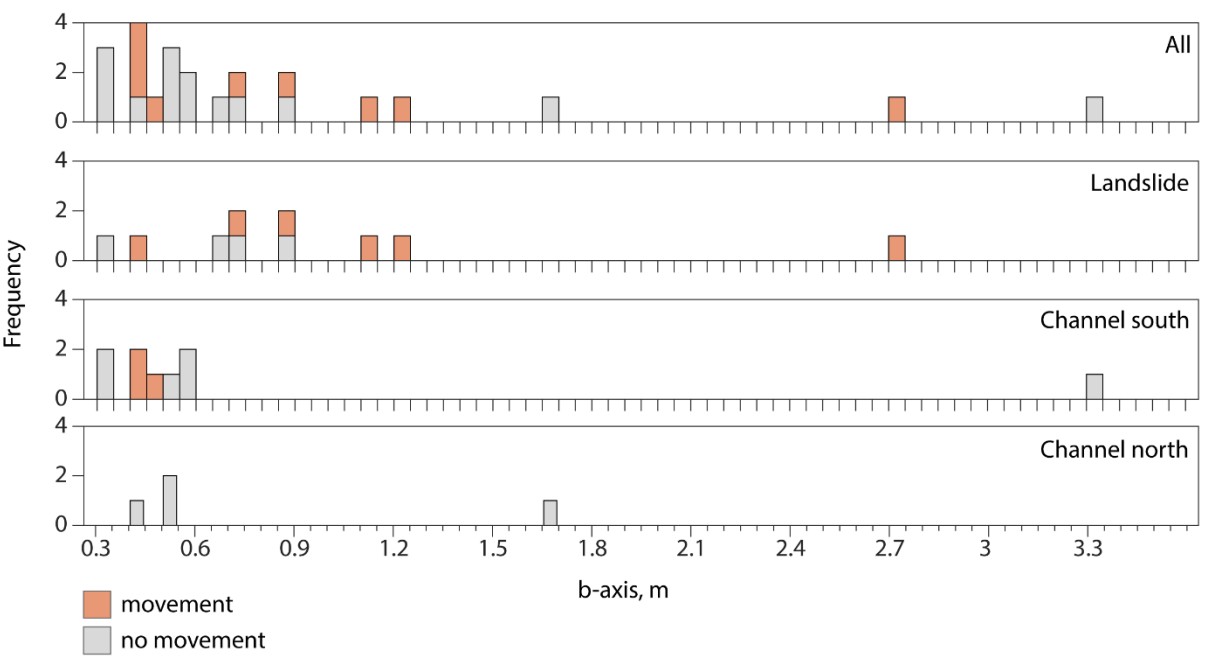

A1. Histograms of boulders b-axis. Colours indicate boulders with movement (light red) or no movement (grey), whilst the panels, top to bottom represent all boulders, landslide boulders, and boulders in the south and north channels respectively. Boulders within the landslide show movements even when their sizes are large, whilst those in the southern channel had preferentially b-axis between 0.4 and 0.5 m.

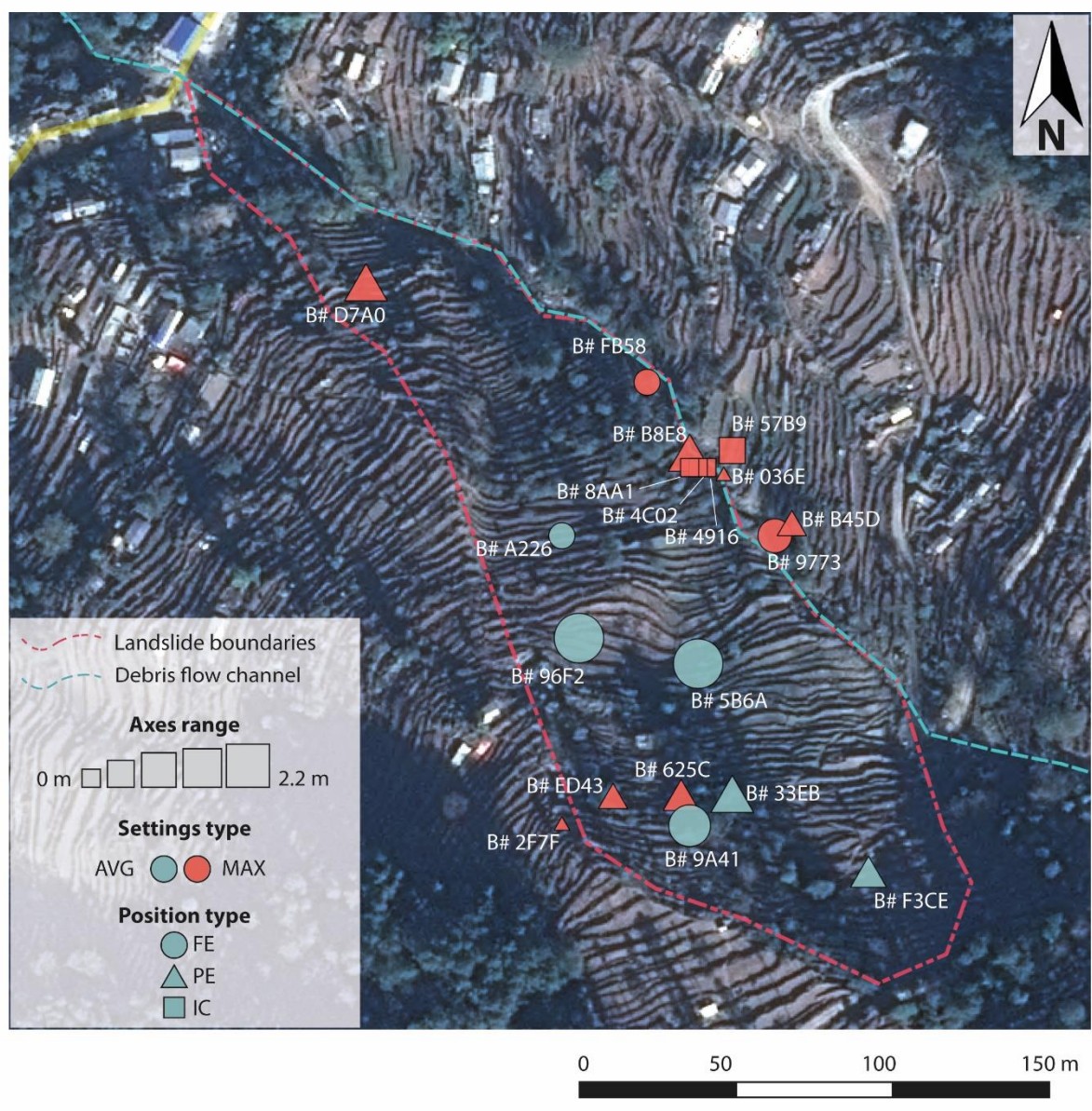

A2. Zoom of two tagged sites. Sizes represent the range between the a-axis and the c-axis of the boulders (equal axes, range 0; most elongated boulders, range 2.2). Sizes are shown in legend for squared symbols but apply to all boulders. Colours represent setting types and symbols represent location type. Image: Pleiades (CEOS Landslides Pilot).

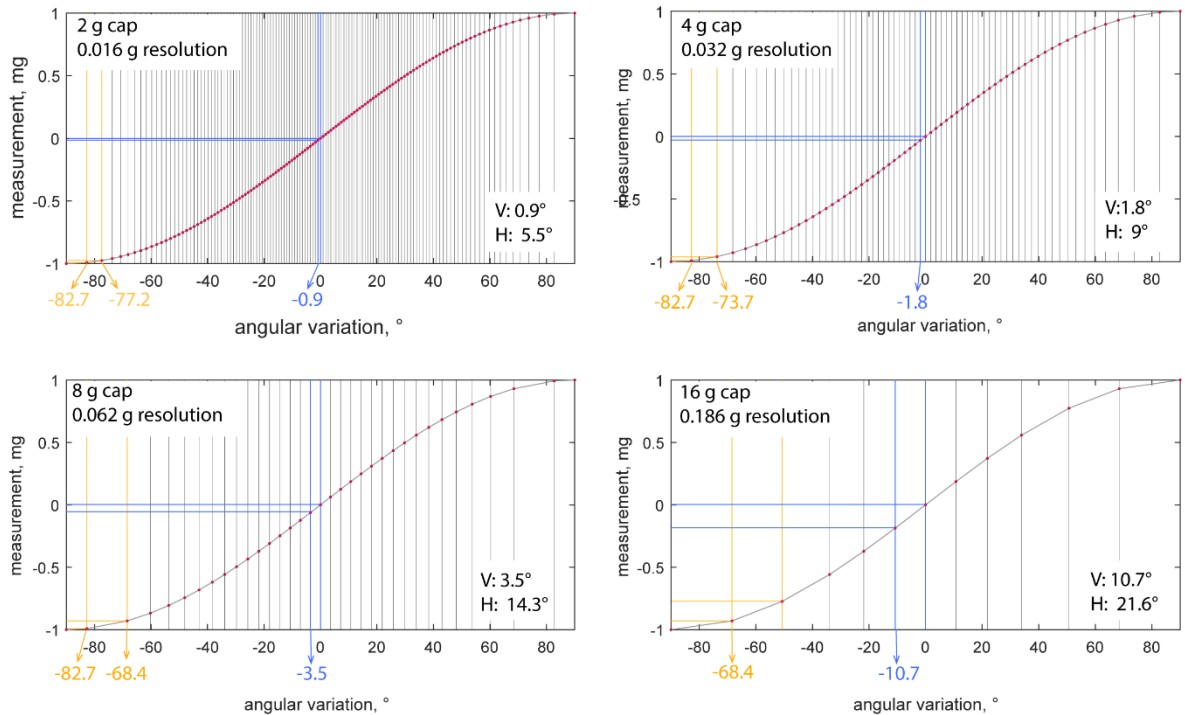

A3. Resolution and sensitivity of the accelerometer with scale capped at 2, 4, 8, and 16 *g* respectively. The vertical lines represent the angular variation corresponding to each step in the scale (mg). The graphs show that for increasing maximum detectable value, the resolution decreases significantly. Moreover, the sensitivity is higher when the axis is vertical than when the axis is horizontal, i.e. when the axis is near horizontal, a larger angular variation is required to make one step in the *g* scale. Thus, the angular threshold used to trigger a fix has to be higher than the maximum angular change needed to make a step in the *g* scale when the axis is near horizontal. This is shown as H in the text box in the plots.