# Peer review of "Development of smart boulders to monitor mass movements via the Internet of Things: A pilot study in Nepal"

_Earth Surface Dynamics, 2020_

## Referee Comment (RC1) · Georgios Maniatis (Referee) · 14 Nov 2020

**Review *esurf-2020-78**

Development of smart boulders to monitor mass movements via the
Internet of Things: A pilot study in Nepal
Authors: Benedetta Dini, Georgina L. Bennett, Aldina M. A. Franco,
Michael R. Z. Whitworth, Kristen L. Cook, Andreas Senn, and John M.
Reynolds

**Summary**

The manuscript by Dini et al, presents a new system for monitoring boulder motion
in landslide dominated environments. The work is presented as a case study exam-
ple, with the system deployed in the very active area of Bhote Koshi catchment north-
east of Kathmandu. The system consists of: a) a stack of multi-sensors, which com-
prise inertial-accelerometers and GPS sensors, and; b) a local LoRa network which is
responsible for the wireless collection of the data and their transmission to the GSM
network. This configuration demonstrates the potential of continuous monitoring of
boulder dynamics and their tracking during a landslide event remotely, from a laptop
or a mobile device connected to the internet.

    The paper:

- introduces the problem of monitoring landslides it its relationship with moni-
  toring individual boulders

- describes the proposed monitoring system at a high level

- introduces the monitoring area focusing on its high vulnerability to landslides

- discusses the processing and filtering of the accelerometer data

- compares the derived acceleration measurements with complementary mea-
  surements of TLS and rainfall data

- evaluates the performance of the sensing system using qualitative comparisons
  between the observed motions (from the accelerometer) and the complemen-
  tary measurements.

    It is important to note that the monitored boulders range in size and were placed
in two neighbouring but different areas: one representative of slowly moving land-
slides and one faster, debris flow controlled channel. The boulders were also placed
at different positions within the landslide (exposed, partially embedded and fully
embedded).

**Overall Evaluation**

This is a truly amazing effort. I know first hand that the IMU technology (accelerometers) is not mature enough for long-term unattended monitoring. The fact that the authors managed to collect data in a "close to real time" manner and demonstrate the use of this technology and its potential to co-exist with both GPS tracking (even if it didn't always work) and the Internet of Things, is remarkable. For this reason, I think that this manuscript can be very useful to the EarthSurfD audience and I want to see it published. At the same time, I have two points of criticism that are, in my opinion, major.

The first is that the paper is too long at places and looses its focus. I believe that the main contribution is the introduction of the sensing system. A little bit less context will benefit the manuscript and allow the reader to focus on the technical aspects of the deployment which are the most difficult (and the most controversial given the maturity of the deployed technology).

The second is relevant to the presentation of the accelerometer data. The authors use a rotation convention which can work in certain electronic engineering applications (when we want to rotate the screen of a smartphone for example), but it is not very relevant to the 3D rotation of the boulder. This is not a matter of semantics, it is important to describe the accelerometer data in more relevant context, even if the focus is not on measuring the actual dynamics but extracting more "qualitative" results or "binary" states (mobility non-mobility, rotation - no-rotation, fast- slow rotation etc.). The authors do not claim that they measure the dynamics accurately (which is very correct), but the way the data are processed and presented makes them very difficult to understand and (more importantly) reproduce, even in isolated laboratory conditions.

I have organised the rest of the document in three sections. The first two are devoted to the problem of analysing the accelerometer data. I attempt to explain where is my main objection and I propose a framework which the authors can use to both shorten the analysis and make it more comprehensive. The last section consists of specific comments on the manuscript.

A disclaimer is needed: I don't claim that what I propose here is the best possible framework to analyse accelerometer data. I only claim that it is probably the most useful given that we are discussing accelerometer measurements only (instead of a full IMU) and that the audience of the journal is not necessarily familiar with the details of this technology yet.

**Analysis of the accelerometer data**

The sections of the manuscript that refer to the calculation of the rotation angles are not referenced well. My guess is that the authors followed this technical note from

NXP: https://www.nxp.com/docs/en/application-note/AN3461.pdf

which is very useful for the people that make embedded systems but (like most of this type of notes) omits a lot of the necessary theory.

My first objection is that the authors calculate the rotations without any information about the initial orientation of the accelerometer sensor. When the manuscript refers to "close to vertical" or "close to horizontal" rotations it is necessary to specify both the frame of reference (horizontal according to what? the global inertial frame?), and which is the accelerometer axis that approaches that level (horizontal or vertical). The fact that the authors moved the time-series around during the plotting, makes this initial orientation even more difficult to understand. In short, there is no guarantee that the accelerometer in the boulder is orientated as shown in figure 5a. As a result, it is not possible to verify the "horizontal" or the "vertical" without calibration *in situ*.

My second objection is that the authors discuss the increased error and the coarser resolution of the axes rotation close to the "horizontal" level, without a clear description of what this is and why it happens. More importantly, it is presented as a sensor/ programming issue which is missleading. The reason the error increases, is called in the theory of rotations "Gimbal Lock". It is the result of the rotations described in the NXP note above as non-commutative, which in plain language means that they are not independent (when one axis changes, the other two change too). There are a lot of useful references for this, one of the most concise and simplified can be found here: http://www.chrobotics.com/library/understanding-euler-angles

My third and final objection has to do with the attempt to record linear accelerations without compensating for gravity. The accelerometer measures the difference between the gravity field and any applied linear acceleration. When the sensor is static it measures any rotation that is not directly aligned with the gravity field. The problem begins when the sensor starts moving (when the linear acceleration is applied). If there is no accurate description of the relative orientation of the sensor with the gravity field (in 3D) available, the two measurements (the static and the "mobile") cannot be decoupled.

To summarise, the presented analysis doesn't offer a true representation of the boulders' movement, despite the fact that 3D accelerometer data are presented. And by "true" I don't imply a fully accurate measurement of the dynamics. The presentation of the data in the manuscript does not allow for a confident observation of the mode of motion (rotation or linear motion), which is critical. This results to a qualitative interpretation of the data which is better than nothing, but a) doesn't explore the full potential of the technology and b) skews even more the already tangled references on the use of accelerometers in the field of geomorphology.

**Proposed framework**

I strongly believe that there is no real need for a 3D (or even 2D) description of the rotations to make this application successful. It is possible to derive metrics that represent the magnitude of rotational changes and the magnitude of the applied linear acceleration without analysing 3D rotations in their full detail. The direction of rotations is not important in the context of early detection. Robust motion detection can be achieved by calculating the "unit quaternion" and by compensating for gravity using the norm of the raw (non-normalised) accelerometer data only.

The quaternions are a group of complex numbers. They have a long history, but they are in the spotlight at the moment because they simplify the rotations of IMUs. There is incredibly large number of references online and ever more guides to implement them in IMU rotations. However, the vast majority of them assumes the presence of a gyroscope which is not available here. For this application, it is enough to treat quaternions as 4-element vectors.

Valenti at al. (https://www.mdpi.com/1424-8220/15/8/19302) provide a solution for an auxiliary quaternion as a part of an optimised sensor fusion which includes a gyroscope and a magnetometer. This solution is auxiliary because it rotates the acceleration vector to the global (earth) horizontal plane, but doesn't define the orientation in 3D (a magnetometer is necessary for that). However, it provides a global "rotation" metric and it avoids singularities, which is the main issue with the convention followed in this manuscript. The authors can refer to the equation 25 of Valenti et. al, which is the following:

$$
{}_i^b q = \begin{cases} \left[ \sqrt{\frac{a_{bz}+1}{2}} \quad -\frac{a_{by}}{\sqrt{2(a_{bz}+1)}} \quad \frac{a_{bx}}{\sqrt{2(a_{bz}+1)}} \quad 0 \right]^\mathsf{T} , a_{bz} \geqslant 0 \\ \left[ -\frac{a_{by}}{\sqrt{2(1-a_{bz})}} \quad \sqrt{\frac{1-a_{bz}}{2}} \quad 0 \quad \frac{a_{bx}}{\sqrt{2(1-a_{bz})}} \right]^\mathsf{T} , a_{bz} < 0 \end{cases}
\tag{1}
$$

where ${}_i^b q$ is the quaternion (a 4-element vector for this application), that rotates the accelerometer from to the global horizontal frame, and $a_{bx}, a_{by}, a_{bz}$ are normalised accelerometer measurements.

If the authors derive a ${}_i^b q$ for each one of the boulders, then the norm of this quaternion is calculated using the following equation:

$$
n(q) = \sqrt{q_2^2 + q_1^2 + q_3^2 + q_4^2}
\tag{2}
$$

where $q_1, q_2, q_3, q_4$ are quaternion elements. I state this calculation here because most of the quaternion operations are different to the typical vector ones. The $n(q)$ of each boulder can be used to check the stability of equation 1 (this is a unit quaternion, the norm should be approximately equal to 1). The ${}_i^b q$ of each boulder is an

unambiguous metric of orientation change. A time-series of $_i^b q$ will show when the boulders have rotated. And the component derivatives of $_i^b q$ can give a quantified metric of the speed of this rotation.

As for the linear acceleration, it is enough to just subtract the gravity norm from the raw acceleration norm. It will be much easier to identify major impacts and understand the noise threshold of the accelerometer when it is static. The manuscript states that the sensor at "maximum" settings did not record anything larger than 1g. If that is the total acceleration and there is no compensation for gravity, then the boulders did not move (according to the accelerometer), or the data did not transmit at all during transport events for the reason I discuss in my third objection on the analysis of the accelerometer data.

My final suggestion is to try this framework (or at least try to extract global metrics or norms instead of accelerometer axes metics) and then evaluate the use of the moving average filters applied in this work. Most of the IMU errors are not linear, but the reasoning of the filtering methodology used in the manuscript is not unfounded. I think that after the calculation of more global or normalised metrics, many spurious values will be punished or clearly identified in the noise threshold of the sensor which makes them easy to remove.

**Specific comments on the manuscript**

Line 21 ".. and sudden rotations"

Those are difficult to distinguish using the accelerometer measurement only.

Line 21, end of paragraph discussing RFID tags

A general note here is that all those techniques work in a "before/ after" event manner. Not suitable for warning.

Line 91 "Recently, the use of IMUs (Inertial Measurement Unit) has been tested for different applications in 91 the field of geomorphology (e.g. Caviezel et al., 2018 and references therein)."

If the scope here is to refer to previous IMU deployments in geomorphology, I would argue that the first one was from Akeila et al. 2010. And the first milestone from Frank et al., 2014. Those refer to fluvial and coastal transport respectively, and the implementations are quite detailed.

Line 95 "..to reconstruct the path and movement of individual particles in a laboratory flume"

This is not accurate. Unfortunately, it is not possible to reconstruct the path of a stone using a standalone IMU.

Line 109 "...accuracy and precision of the measurement itself, the latter requiring further development"

Very important comment this one. I sincerely appreciate it.

Line 116 "... the energy thresholds required for remobilisation of different grain sizes"

The term "energy" here is quite misleading. Is this a reference to kinetic energy? I think re-wording this to "forcing" will clarify this sentence

Lines 132-138

There are paragraphs like this that they need to be summarised more. I know the advantage of a wireless semi-automated warning system is clear. I think that there is a little bit more space on arguing about the benefits of the deployed system that it is required.

Line 142 "We also demonstrate for the first time the use of this technology in the field of geomorphology, and in a field setup, to monitor the movement of boulders embedded within a landslide and in two debris flow channels."

That is a very strong statement. There are few applications of IMU sensors in fluvial environments. Unless the authors mean the wireless transmission of data through the local network, which is, to my knowledge, a widely applied technology for environmental studies too.
I think that the sections 2.2 and 2.3 can be summarised. I understand the need to describe the site, but if this paper is more about the deployment of the system, then the focus must be on the method and its validation. This amount of background information just shifts the focus, in my opinion at least.

Line 249 " The sensors are equipped with an accelerometer configured to sample at 2 Hz, as well as a GPS module"

The first question that pops up to mind is " is that enough?" I think we are talking about very gradual motion (before the long intense event). A little bit more discussion about the sampling frequency is necessary.

Line 250 "When movement is detected by the accelerometer, so that tilt or acceleration exceed defined thresholds,.."

This means that the sensor is programmed in a "sleep - wake" routine. What is the frequency of the measurement for the "sleep" state? I assume that the 2Hz sampling frequency refers to the "active" state.

Line 261 "The depth of the hole allowed for the emplacement of the C-cell batteries and the sensor. After placement, each hole was filled with epoxy resin, sealing the cavity, thus protecting the device from tampering and from the elements..."

Is there any consideration regarding the orientation of the sensor in respect to the frame (the 3D volume) of the boulder?

Line 277 "The devices deployed in the 2019 season were programmed to not store the data, but to send it immediately, causing the data transmitted during gateway offline time to be lost."

Very common issue, I have been there and it is a very hard lesson to learn. It hurts even if you can repeat the experiment in the next hour.

Line 304: "... with an accelerometer event for which activation thresholds can be set for impact forces and for angular variations."

Detail is needed for this threshold

Line 309: "In the first case, the values of the three axes are normalised and the measurements essentially represent the static angle of tilt or inclination, thus the projection of the acceleration of gravity, g, on the three axes, ranging between 0 (for a horizontal axis) and $\pm 1$ (for a vertical axis)."

Question a: normalised with the acceleration norm I guess, needs to be clarified.
Question b: how the normalised measurement is a direct measure of the tilt angle? Many references are needed here.

Line 321 "...and m is accelerometer value recorded on the same axis in g"

Is this a normalised measurement?

Line 322 "..  = 0.016 g the corresponding angular variation is of  0.9 if the axis is vertical, but  5.5 if the axis approaches horizontal"

I think the authors use the normalised accelerations and only positive angle 2D changes. This is a very small subcategory of 3D rotations. In addition, it is necessary to describe more equation 1. And the authors need to explain why the resolution changes according the initial orientation of the sensor.

Line 327 "...variability in accelerometer measurements around a stable value, rather than true movement, with this effect becoming more important in sensors programmed with the coarser scale."

The way this is written, it implies that the error and the scaling are programming/sensor issues. They are not. The differences in scaling appear because the authors used Euler angles (yaw and pitch from what I can understand) which result into singularities. This could be avoided with the use of quaternions.

Line 339 "Measurement variability and errors related to the sensors led to spurious data, given the relatively small angular threshold assigned for the highest detectable maximum of 16 g. In other words, given that the step of accelerometer measurement is as high as 0.186 g, a spurious angular variation of more than 5 is often detected even when the boulder is stable, due to intrinsic measurement variability (up to 2 bits)."

This may be an artefact of calculating subsequent orientations and the integration of those. It is very difficult to tell from this presentation.

Line 346 "within $\pm$ 0.186 g of the data point immediately before the window. If any of the values lie outside the $\pm$ 0.186 g threshold.."

This averaging corresponds to the "near vertical scenario". What if the sensor is on the "near horizontal" state?

Line 350 "This would mean that a high value would likely be followed by a change in the static angle of tilt of the three axes"

Here, the case of a linear motion without rotation is not captured.

Line 351 "Therefore, it is unrealistic to have a peak value followed by a value equal to that observed before the peak."

If the sampling frequency is at 2Hz, this is potentially true for the larger boulders. But not safe to assume for the smaller ones.

Line 359 "The accelerometer readout in the current version of the software is tied to a GPS acquisition, this means that although the accelerometer is measuring as soon as movement is detected, the acquisition is obtained only when the GPS has successfully retrieved the position."

This is not an issue, as long as there is a clear description of when the movement is detected (time and acceleration threshold)

This is not an issue, as long as there is a clear description of when the movement is detected (time and acceleration threshold for the "wake up" of the sensor.

A quantification of how "rough" is this estimate here would be useful.

Line 386 This data is used in a qualitative way for comparison with and validation of the accelerometer data obtained with the wireless devices and, despite the qualitative approach, this data provided a quite detailed overview of the days in which movement occurred.

It is important to stay here how the authors associate the geomorphic change with the accelerometer data. It is necessary to recognise that the comparison refers to two vastly different time resolutions.

Line 421: "The values of each axis are recalculated to show the deviation from the original position for visualisation purposes, rather than the actual values measured (hence all raw data curves begin at 0, and the smoothed curves around zero, due to the smoothing)."

This can be very misleading. The initial orientation is crucial for interpreting the accelerometer data.

Line 434 "Fig. 4 and 5 show that the movements of boulders within the landslide not only differ in the magnitude of the angular variations recorded, which is an order of magnitude higher for B A226 and B 9A41 in comparison to other boulders, but also in the evolution with time."

There needs to be an objective, quantified metric for this comparison. There is plenty of data.

Line 461 "These boulders were programmed to retrieve actual g values (as opposed to normalised values) and forces up to 16 g."

This needs be highlighted and clarified much earlier in the manuscript

Line 473 "We do not observe forces > 1 g for any of the sensors programmed with the maximum settings, despite the ability of the sensors to detect up to 16 g. This is consistent with a lack of debris flow activity recorded by cameras or seismometers, the more prolonged activity of which would have generated sustained boulder movement, beyond the time needed for GPS acquisition as explained below"

This is not compatible with the detection of linear accelerations. Compensation for gravity is required.

Lines 458-459 require further quantification

Lines 508-515

I know that those deployments are extremely difficult and they don't always go to plan. But the discussion of the GPS data here is not very useful. It is both too long and not directly feeding to the interpretation of the data. My honest opinion is that the paper would benefit if the GPS data were not discussed in the main body. It is probably useful for an appendix to demonstrate the deployment, but there is no clear quantitative information that can be extracted from here.

Line 543 "The movements observed for the boulders scattered on the landslide body and embedded within the material can be described as small angular variations that occurred gradually during the season"

Those are the type of statements that require further quantification.

Line 550 "...show higher magnitude of the angular variations with respect to other boulders (Fig. 5F-G)."

How much higher?

Line 552 "..follows a spatial and temporal pattern"

This is a very strong statement. I would definitively require some statistical justification.

Line 588 "What controls this behaviour is not the fact that the sensors were programmed to detect the maximum force or the static tilt respectively, but rather the scale that was chosen and associated with the two settings types."

That can be true, but it is not the only reason for increased noise. My first guess would be de-callibration or humidity. Those sensors are very temperamental. And most of the noise is traditional, random AC-DC circuit noise.

Line 602 "Future improvements of the accelerometer accuracy, resulting for example from the activation of the 9-axes IMU present in the hardware of the devices, could reduce this problem"

This is very true. Especially the gyroscope measurement will be very useful for this type of measurement.

Line 616 " The high positional errors and the important battery expenditure make the current GPS module not fit for the purpose of tracking boulders in rugged terrains."

That is also very true. We need to investigate or come up with alternative tracking techniques for remote areas.

Line 630 " This may explain why, although the boulders in the channel were programmed to detect high forces, they never show accelerometer values higher than 1 g (either negative or positive)."

This is true, but I think there is also an artefact of the processing followed here. I am not sure the authors will pick up very high inertial force. The boulders are quite large and heavy. Maybe 1g is too small though (see notes above).

The last section of the discussion (5.1) is useful but I think it could be summarised a lot. A table of prons and cons would be a good addition.

Line 730 "...but that in the future are expected to replace the need for an accurate GPS."

I would strongly oppose that. There is no evidence that the available IMUs will be suitable for standalone tracking anytime soon. Unless, the authors refer to military grade optical sensors which cost £10k each. If that is the case it is necessary to provide some specs.

GM

---

## Referee Comment (RC3) · Anonymous Referee #2 · 7 Dec 2020

The presented work focuses on the deployment of accelerometers and its real-time data transmission as possible low cost means of surveillance for large single blocks to identifiy mass movement associated with landslide type of rock slides. It presents a substantial an thoroughly carried out field measurement campaign and careful data analysis. The use of IMU technology for boulder tracking and its possible applications for early warning systems is a highly relevant topic. The comprehensive presentation of this pilot study definitevly merits publication after some minor revisions.

Generally, the presentation of the entire work is very nicely done. I also have to thank the authors to present a carefully edited and proofread manuscript, which made read-

ing easy and enjoyable.

In the following some content and technical suggestions for improvement and additional context are provided. IMHO the manuscript would benefit of some remarks on remaining challenges and disadvantages of IMU tracking/signal processing.

Introduction l41: Large boulder movement rarely comes isolated. While the approach to use large boulders as particle marker for mass movements with modern technology is new, the general statement that the motion of large boulders and its damage potential is not discussed in literature may be a bit exaggerated.

L54: large boulders can be detected via RADAR/LiDAR technology, which is truly remote. The target boulders here predominantly are early warning signs

L64: State-of-the art RADAR (no interferometric RADAR of course) techniques are able to deliver real-time data for immediate mitigation actions such as road closures etc. See https://ui.adsabs.harvard.edu/abs/2020EGUGA..22.5138W/abstract for the lack of better reference sake.

General remark: With all the advantages listed for the IMU technology applied, one crucial disadvantage needs to me mentioned: The installation of the sensors do require physical presence at the block. While this may not be a problem for large boulder instrumentation in slowly evolving mass movements, this is certainly a major drawback to deploy the presented technique in active sites.

Methodology 3.1 Network setup and components Really nicely presented methodology!

Notation remarks: • Generally throughout the manuscript, change the notation of the local gravitational field of Earth to \texit{g} or $g$ as it denotes a physical constant usually denoted in italic font. This also removes the ambiguity of mg and mg. • The same holds for x,y and z axis, variables denoted by italic characters. Any given coordinate system is given by its n-space.

3.2 Choice of tracked boulders l298 coherently → collectively/mutually. Coherence would imply that the motion pattern is the same, as a laser has coherent wavelengths. Large boulders can move with the landslide but usually succumb to a slightly different kinematical regime. True coherence in nature is extremely rare.

3.3 Sensor Settings l323ff replace the "∼" with \approx or the word roughly, about, etc.. Tilde means "similar to" and is usually used in plain mathematical context.

l352 maybe add "before the peak when sampled at 2 Hz." If sampled at higher frequency, such double or three peak hits are not that uncommon.

4 Result Thorough presentation of the results. Only notation of axis and g and "∼" characters would need some attention.

5 Discussion Validation of motion is partly done via camera imagery. While I would agree that only tilting motion of an embedded rock is not feasible to be detected via imagery, I would argue with the progress in resolution an image processing, a pixel tracking via cross correlation analysis of interval imagery might well track slow motion onsets. The spatial resolution is then given by the camera's resolution. Just one of a zoo of cross-correlation papers (https://nhess.copernicus.org/articles/17/2143/2017/ )

L668 while in the introduction the heritage of animal tracking is mentioned, a comparison with state of the art logistic tracking devices such as MSR sensors or trusted global devices (just to name two), would be interesting. Modern logistic shock tracker do also work with acceleration and angular velocity IMUs and sometimes even come with satellite network coverage to send the reports.

L688 As stated by the authors, independce of GPS/GNSS signals is of paramount importance.

L731 Accurate position information from IMU sensor integration requires sophisticated post-processing procedures in order to minimize integration error accumulation. This is feasible in case of periodic motion or motion patterns, where at specific positions in

time a zeroing of the errors is possible. If this is not the case, accurate position tracking via IMU is extremely challenging, especially for fast motion. If GNSS (maybe refer to GNSS than GPS alone, as there are many other systems in the sky then GPS only) measurements will become obsolete in the future, one will see.

---

## Author Comment (AC1) · 15 Jan 2021

Dear Dr Maniatis,

thank you very much for your comprehensive, clear and structured review. We found this very helpful and it made us reflect upon how to improve not only the paper but also our understanding of the problem we tackled. We also thank you for recognising the effort that went into setting up the network, collecting and analysing this data for the first time in a real, field environment.

We strived to address your comments and made the necessary clarifications throughout the manuscript accordingly. Please refer to the pdf attached, in which we answer each of your comments one by one.

Benedetta Dini

Please also note the supplement to this comment:
https://esurf.copernicus.org/preprints/esurf-2020-78/esurf-2020-78-AC1-supplement.pdf

---

## Author Response (AR1)

Reviewer #1

Review *esurf-2020-78*

Development of smart boulders to monitor mass movements via the Internet of Things: A pilot study in Nepal
Authors: Benedetta Dini, Georgina L. Bennett, Aldina M. A. Franco, Michael R. Z. Whitworth, Kristen L. Cook, Andreas Senn, and John M. Reynolds

**Summary**

The manuscript by Dini et al, presents a new system for monitoring boulder motion in landslide dominated environments. The work is presented as a case study exam- ple, with the system deployed in the very active area of Bhote Koshi catchment north- east of Kathmandu. The system consists of: a) a stack of multi-sensors, which com- prise inertial-accelerometers and GPS sensors, and; b) a local LoRa network which is responsible for the wireless collection of the data and their transmission to the GSM network. This configuration demonstrates the potential of continuous monitoring of boulder dynamics and their tracking during a landslide event remotely, from a laptop or a mobile device connected to the internet.

The paper:

- introduces the problem of monitoring landslides it its relationship with moni- toring individual boulders

- describes the proposed monitoring system at a high level

- introduces the monitoring area focusing on its high vulnerability to landslides

- discusses the processing and filtering of the accelerometer data

- compares the derived acceleration measurements with complementary mea- surements of TLS and rainfall data

- evaluates the performance of the sensing system using qualitative comparisons between the observed motions (from the accelerometer) and the complemen- tary measurements.

It is important to note that the monitored boulders range in size and were placed in two neighbouring but different areas: one representative of slowly moving land- slides and one faster, debris flow controlled channel. The boulders were also placed at different positions within the landslide (exposed, partially embedded and fully embedded).

**Overall Evaluation**

This is a truly amazing effort. I know first hand that the IMU technology (accelerometers) is not mature enough for long-term unattended monitoring. The fact that the authors managed to collect data in a "close to real time" manner and demonstrate the use of this technology and its potential to co-exist with both GPS tracking (even if it didn't always work) and the Internet of Things, is remarkable. For this reason, I think that this manuscript can be very useful to the EarthSurfD audience and I want to see it published. At the same time, I have two points of criticism that are, in my opinion, major.

Dear Dr Maniatis, thank you very much for your comprehensive, clear and structured review. We found this very helpful and it made us reflect upon how to improve not only the paper but also our understanding of the problem we tackled. We also thank you for recognising the effort that went into setting up the network, collecting and analysing this data for the first time in a real, field environment.

We strived to address your comments and made the necessary clarifications throughout the manuscript accordingly. We answer each of your comments one by one for clarity in teal below.

The first is that the paper is too long at places and looses its focus. I believe that the main contribution is the introduction of the sensing system. A little bit less context will benefit the manuscript and allow the reader to focus on the technical aspects of the deployment which are the most difficult (and the most controversial given the maturity of the deployed technology).

We fully appreciate this comment and we have already tried to keep the manuscript within reasonable length during the first drafting. However, we would like to give a comprehensive background, from the point of view of the geomorphology (to leave the motivation clear) and the technology (which may be new to some readers of Earth Surface Dynamics). We also think it is good practice to contextualise our chosen technology in the panorama of other technologies used in overlapping fields of research. We think it is beneficial to leave the content as it is. Hopefully the reader is aided by a clear structure.

The second is relevant to the presentation of the accelerometer data. The authors use a rotation convention which can work in certain electronic engineering applications (when we want to rotate the screen of a smartphone for example), but it is not very relevant to the 3D rotation of the boulder. This is not a matter of semantics, it is important to describe the accelerometer data in more relevant context, even if the focus is not on measuring the actual dynamics but extracting more "qualitative" results or "binary" states (mobility non-mobility, rotation - no-rotation, fast- slow rotation etc.). The authors do not claim that they measure the dynamics accurately (which is very correct), but the way the data are processed and presented makes them very difficult to understand and (more importantly) reproduce, even in isolated laboratory conditions.

The choice of how to represent the data was certainly not an immediate one. As you correctly point out, the nature of the data we deal with here makes it more familiar to scientists close to the engineering realm than to those who work in the field of geomorphology. This is why we prefer to present the (almost) raw accelerometer data, presenting plots in which either gradual changes or "jumps" in the x, y and z axes values are immediately visible. We gather you understood that we did calculations to obtain 3D orientations to show in our plots in Fig. 5 and 6, but this is not the case. We attempted to clarify this even further.
Moreover, we decided to associated to the accelerometer data a simple "schematic" of how the change in orientation of the boulders monitored *might have* looked like, and perhaps this is what has generated confusion. We have no ambition here to claim that such schematics (i.e. those in figure 5A,B and 6B,D,F) are a real representation of our boulders.

First of all, to answer to the comment that the data is hard to reproduce, we would

argue that this is not the case, since we were simply distinguishing binary states (mobility and non-mobility). We also added schematics that may help interpret similar boulder movements. The ability to reproduce similar boulder movement is something that is out of our control, since, as we explain in the manuscript, we do not use an experimental setup but aim at capturing naturally occurring movements. Regarding the reproducibility of our data display, in our methodology section we explain that:

1) we plotted the raw data simply as the difference (in g$^{-3}$) between each data point and the initial value for each axis. So, the data is not changed, but only referred to an initial "zero" state, merely to simplify the look of the graphs; original l.421-422 stated: "The values of each axis are recalculated to show the deviation from the original position for visualisation purposes, rather than the actual values measured".
2) we removed some known noise from the data by removing peaks on a moving window, as detailed in original l. 344-355. The aim of this is to remove spurious peaks that fluctuate around a given value ±1 or ±2 steps of the chosen scale. We know that our devices are likely to show measurement variability of ±1 or ±2 steps.

Second, to address the comment about the 3D visualisation, the boulder sketches we use for our examples in figures 5A, B and 6B, D, F are just model boulders. Figure 5(A,B) only indicate a *possible* situation in which a subtle boulder movement is caused by the landslide's movement, whereas in figure 6(B,D,F) the model boulder floats in a space with no coordinates and with no reference to a real slope. The only reason to introduce such visualisation is to give a sense to the readers of what the change in orientations suggested by our raw data shown in the plots *may have looked like* in a physical space. We clarified this in the text, more precisely in the captions of figures 5 and 6 and current l. 397-404.

For the representations shown in figure 6(B,D,F), we use the following equations, in order to calculate the change in orientation that must have occurred between **two successive static measurements**:

$$\vartheta = \tan^{-1} \frac{Acc_x}{\sqrt[2]{Acc_y{}^2 + Acc_z{}^2}} \qquad (1)$$

$$\varphi = \tan^{-1} \frac{Acc_y}{\sqrt[2]{Acc_x{}^2 + Acc_z{}^2}} \qquad (2)$$

where, $\vartheta$ and $\varphi$ are the pitch and roll angle respectively (as you correctly understood), in a Euler system, as explained in the reference you suggested (http://www.chrobotics.com/library/understanding-euler-angles) and in others sources (e.g. https://www.settorezero.com/wordpress/cosa-sono-come-funzionano-e-a-cosa-servono-gli-accelerometri/).
We know the initial x, y, z measurements, thus, we think it is justifiable to calculate the 3D rotation in terms of pitch and roll angle. The only information we do not have is the rotation around the true vertical (in the Earth's reference system), because this cannot be calculated with accelerometer data only and, moreover, we don't know what cardinal direction our sensors were facing. This is why we do not show the actual orientation of the devices with respect to the real slope.

I have organised the rest of the document in three sections. The first two are devoted to the problem of analysing the accelerometer data. I attempt to explain where is my main objection and I propose a framework which the authors can use to both shorten the analysis and make it more comprehensive. The last section consists of specific comments on the

manuscript.

A disclaimer is needed: I don't claim that what I propose here is the best possible framework to analyse accelerometer data. I only claim that it is probably the most useful given that we are discussing accelerometer measurements only (instead of a full IMU) and that the audience of the journal is not necessarily familiar with the details of this technology yet.

We considered this a good suggestion, and attempted to use the quaternions framework and apply it to our data. However, after testing this method on our data, we think that the framework of the quaternions will not help the readers (those not familiar with quaternions) to more quickly understand the representation of the data and, most of all, the meaning of it, for the purpose of capturing initiation of boulder movements. Below we hope to clarify why.

**Analysis of the accelerometer data**

The sections of the manuscript that refer to the calculation of the rotation angles are not referenced well. My guess is that the authors followed this technical note from

NXP: https://www.nxp.com/docs/en/application-note/AN3461.pdf

which is very useful for the people that make embedded systems but (like most of this type of notes) omits a lot of the necessary theory.

My first objection is that the authors calculate the rotations without any information about the initial orientation of the accelerometer sensor. When the manuscript refers to "close to vertical" or "close to horizontal" rotations it is necessary to specify both the frame of reference (horizontal according to what? the global inertial frame?), and which is the accelerometer axis that approaches that level (horizontal or vertical).

First of all, the identification of movement does not happen on any calculated rotation but on the (almost) raw x, y, z data (see original l: 420-423 "whilst the y axis indicates the value of the projection of g on each accelerometer axis in mg (g-3). The values of each axis are recalculated to show the deviation from the original position for visualisation purposes". Moreover, the fact that we calculate the rotations "without any information about the initial orientation of the accelerometer sensor" is not true and we partly already explained this above. The reason why we could plot time series of the accelerometer measurements is that we are in possess of the raw x, y, z data recorded by our accelerometers! Therefore, if for example we have Z=1 and X=0 and Y=0, we know that the device is in a flat position with respect to the Earth's reference frame, with Z vertical and parallel to the vertical of the global inertial frame). What we did not take note of when we embedded the devices in the boulders is, if for example a device is flat (z is vertical), what cardinal direction the x and y axes point toward. This means that we do know indeed the orientation of the device with respect to gravity and with respect to the Earth system's horizontal, moreover we cannot calculate the yaw angle (about the Z). This is the reason why our schematic 3D representations only aim at showing the rotation that may have occurred between a static orientation 1 and a static orientation 2, in terms of pitch and roll angles only. As mentioned above, it can be noted that the model boulders float in a space with no coordinates, because they are not placed in the context of the channel.
We clarified this further at current l.472-480; 520-521.

The fact that the authors moved the time-series around during the plotting, makes this initial orientation even more difficult to understand.

The time series are not "moved around". What we did is simply to calculate the deviation from the initial value (original l. 421-433). I.e. for i=2:n, $x_t = x_i - x_1$, where $x_t$ is the "transformed" plotted value. Hence, if the raw x,y,z values were plotted, they would look exactly the same as the grey lines shown in our plots in figures 5 and 6, but the readability would be a lot more difficult, due to the scale which would stretch between -1000 and 1000. The rationale behind this is that we simply observe the variation in the values in each axis to identify whether movement has occurred or not, thus it does not matter in what position exactly the sensors were at the start. We added some clarification in text, current l. 388-404.

In short, there is no guarantee that the accelerometer in the boulder is orientated as shown in figure 5a. As a result, it is not possible to verify the "horizontal" or the "vertical" without calibration *in situ*.

We think that we have been misinterpreted. As explained above, the boulder shown in figure 5A is only a schematic representation and it is by no means an attempt to show a real monitored boulder or a real position (Original caption read: "Sketch of possible movement" and it has been further clarified to read: "Sketch of possible type of movement experienced by embedded or partly embedded boulders. Note that this is only a schematic to indicate a movement that occurs in accordance with the landslide body and does not necessarily represent real movement of the boulders monitored in this study".). In short, the sketch shows neither a real boulder nor the real slope, but it shows what might be happening to boulders that slowly move with the mass (original l. 417-418) With respect to figure 6(B,D,F) as mentioned above, it is true that we cannot verify the change in position with respect to the Earth's vertical (i.e. yaw), but we disagree about the fact that we cannot verify changes with respect to the Earth horizontal (pitch and roll), because we have the accelerometer readings of the gravity field recorded by our sensors, including a known original orientation with respect to the inertial frame system. The pitch and roll are calculated as shown in equations 1 and 2 above. Clarified further at current l. 397-404, 478-480.

My second objection is that the authors discuss the increased error and the coarser resolution of the axes rotation close to the "horizontal" level, without a clear description of what this is and why it happens. More importantly, it is presented as a sensor/ programming issue which is missleading. The reason the error increases, is called in the theory of rotations "Gimbal Lock". It is the result of the rotations described in the NXP note above as non-commutative, which in plain language means that they are not independent (when one axis changes, the other two change too). There are a lot of useful references for this, one of the most concise and simplified can be found here: http://www.chrobotics.com/library/understanding-euler-angles

We reckon that here different things got mixed up together. We try to address the comments in order.

1) At original l. 313-315 we state: "The measurement resolution changes according to the chosen detectable maximum, so that a scale capped at 2 g has a resolution of 0.016 g, whilst a scale capped at 16 g has a resolution of 0.184 g (Appendix 3)".
In appendix 3, we show why the resolution becomes coarser as the scale programmed in the devices has a higher range. This is related to the fact that even if we would like to capture accelerations at increased ranges (e.g. max 16 g) the architecture of the data and the way the information is packed in a data string does not allow for increased resolution. Hence, if there are minor fluctuations in accelerometer values recorded even if *the device is static,* these might result in fluctuations that correspond to one or sometimes two steps on the scale. The steps are larger for a scale capped at higher values, as shown in appendix 3. It is not an error that increases with time whilst recording, but an error that increases with programming the devices to capture a max of 2 or of 16 g.

2) At no point in the paper we say that this effect is a result of a programming error. Probably this sentence was misleading and badly placed: "As a consequence of the different resolutions, we observed acquisitions of data triggered by small variability in accelerometer measurements around a stable value, rather than true movement, with this effect becoming more important in sensors programmed with the coarser scale." Original l. 588-590: "What controls this behaviour is not the fact that the 588 sensors were programmed to detect the maximum force or the static tilt respectively, but rather the 589 scale that was chosen and associated with the two settings types".

The coarseness of the scale changes with the detectable range we impose. What the programming error actually caused is this: if we impose an angular threshold that is lower than the scale resolution, we will trigger spurious acquisitions. At original l. 339-

343 we state: "Measurement variability and errors related to the sensors led to spurious data, given the relatively small angular threshold assigned for the highest detectable maximum of 16 g. In other words, given that the step of accelerometer measurement is as high as 0.184 g, a spurious angular variation of more than 5° is often detected even when the boulder is stable, due to intrinsic measurement variability (up to 2 bits)". This can be seen in the graphs in Appendix 3.

We believe that this is related to the fact that even if the device is static, the natural variability in the acquisitions will trick the device into sensing an angular change that is not real. The fluctuations would have been observed anyway, in the regular daily acquisitions, but we argue that we triggered more acquisitions because of this effect (remember we have an activation angular threshold). This is an unwanted behaviour, because we want ideally the devices to be triggered by real movement only. So, in this sense, setting the angular threshold lower than the resolution was a programming error. We have attempted to clarify this at current l. 369-372 and current l. 661-662.

3) We don't think we observed increase of error in time due to Gimbal lock. Although it is true that the three axes are not independent, the error we are talking about here is just a fluctuation of the values recorded, due to small variability in the accelerometer recordings, which might correspond to relatively high step changes even for a static device, due to coarser resolution. The error is bound to increase with decreasing resolution, as shown in appendix 3. We do not believe either that we see an increased error in time, if this is what is implied in the comment above, and this can be seen in the graphs in figure 5 and 6 but is also shown in figures 1-5 of this document. Gimbal lock is of no relevance in this case, because we are not really dealing with Euler angles data, but rather the accelerometer data (almost) raw.

My third and final objection has to do with the attempt to record linear accelerations without compensating for gravity. The accelerometer measures the difference between the gravity field and any applied linear acceleration. When the sensor is static it measures any rotation that is not directly aligned with the gravity field. The problem begins when the sensor starts moving (when the linear acceleration is ap- plied). If there is no accurate description of the relative orientation of the sensor with the gravity field (in 3D) available, the two measurements (the static and the "mobile") cannot be decoupled.

We programmed some devices to *be able to detect* high impact forces, which is what you refer to as linear accelerations. But, in this case study, we were never able to do so due to two main factors. 1) We only recorded accelerometer data at 2Hz (we now have increased capability, the newest developments will allow for much higher sampling frequency), thus we might have missed "impacts" due to low sampling; 2) As we explained in detail at original l. 362-368 and then again original l. 625-646.
the GPS took always very long (up to 120 seconds) to acquire a position (or fail), and during this time, any fast movement would have the time to unfold and stop. This is explained in detail in section 4.2 and we clarified it at current l. 414-422. Therefore, we know we have not captured the full movement, but only the change between one orientation and the next. This makes the main point of our work the *detection*, within seconds (real or near-real time), *of a change in orientation* which, we infer, must be caused by a movement, like two snapshots in time. This was already explained at original l. 633 onwards: "In essence, these sensors have also only recorded the static tilt and different orientations acquired by the boulders in time, but not the actual movement as it unfolded. […]". This problem should also be improved with the current developments, as we are now able to separate GPS and accelerometer acquisition and transmission, in favour of a quicker response of the accelerometer, coupled with a higher sampling frequency.
But we do not think that we need to compensate for gravity in this dataset, because what we use is the data of two successive (static) orientations, even if these are acquired only a few seconds apart. At original l.473-477 we explain that we do not observe values >1g, likely because the movement did not continue for longer than the GPS acquisition needed (120

seconds), thus we only observe the projection of gravity on x,y,z between a first orientation and a second orientation attained after the movement occurred. Figure 6 caption reads: "Model boulder 3D visualisation to represent the change from the initial positions of the boulders and the positions acquired after the recorded movement." This is why we refer throughout the paper to "orientation changes".

With accelerometer data only, we cannot resolve full rotational behaviour anyway, which is why further development is focusing on activation of gyroscope as well.

To summarise, the presented analysis doesn't offer a true representation of the boulders' movement, despite the fact that 3D accelerometer data are presented. And by "true" I don't imply a fully accurate measurement of the dynamics. The presentation of the data in the manuscript does not allow for a confident observation of the mode of motion (rotation or linear motion), which is critical. This results to a qualitative interpretation of the data which is better than nothing, but a) doesn't ex- plore the full potential of the technology and b) skews even more the already tangled references on the use of accelerometers in the field of geomorphology.

As explained above, the main point of this paper is not that of offering a 3D representation of boulder movement. The 3D representations are only shown to give a sense of the change in orientation a boulder might have undergone, but the full movement between one orientation and the successive, was not recorded here, due to GPS delay and low sampling frequency.
A representation of the mode of motion is critical, we agree, if the aim is that of understanding the dynamics of the processes observed, which might be attempted in future studies by ourselves thanks to current developments undergoing. However, the main point of this first paper is to show that this technology might mature quickly in the near future to provide real time data on the initiation of hazardous boulder movement, which we believe our data already highlights, despite the capability limitation at the time of data acquisition. We do not imply in our study to have explored the full potential of the technique yet, but rather we show for the first time this technique used in a real, field setup (as opposed to experimental) and we show promising results in detecting onset of boulder movement in a near real time. We also indicate what are the steps we have begun to take after collecting this initial batch of data and we mention the development needed. We do not fully understand what is meant by: "skews even more the already tangled references…". Finally, visualizing x, y, z accelerometer data is quite commonly done when using this type of information, e.g. Caviezel et al., 2018.

**Proposed framework**

I strongly believe that there is no real need for a 3D (or even 2D) description of the rotations to make this application successful. It is possible to derive metrics that represent the magnitude of rotational changes and the magnitude of the applied linear acceleration without analysing 3D rotations in their full detail. The direction of rotations is not important in the context of early detection. Robust motion detection can be achieved by calculating the "unit quaternion" and by compensating for gravity using the norm of the raw (non-normalised) accelerometer data only.

As you correctly point out, direction of rotation is not important for early detection. This is why we work with assigning a rotation threshold for activation of the devices.

The quaternions are a group of complex numbers. They have a long history, but they are in the spotlight at the moment because they simplify the rotations of IMUs. There is incredibly large number of references online and ever more guides to implement them in IMU rotations. However, the vast majority of them assumes the presence of a gyroscope which is not available here. For this application, it is enough to treat quaternions as 4-element vectors.

Valenti at al. (https://www.mdpi.com/1424-8220/15/8/19302) provide a solution for an auxiliary quaternion as a part of an optimised sensor fusion which includes a gyroscope and a magnetometer. This solution is auxiliary because it rotates the acceleration vector to the global (earth) horizontal plane, but doesn't define the orientation in 3D (a magnetometer is necessary for that). However, it provides a global "rotation" metric and it avoids singularities, which is the main issue with the convention followed in this manuscript. The authors can refer to the equation 25 of Valenti et. al, which is the following:

$$
{}^{b}_{i}q = \begin{cases} \left[ \sqrt{\dfrac{a_{bz}+1}{2}} \quad -\dfrac{a_{by}}{\sqrt{2(a_{bz}+1)}} \quad \dfrac{a_{bx}}{\sqrt{2(a_{bz}+1)}} \quad 0 \right]^{T} , a_{bz} \geqslant 0 \\[2em] \left[ -\dfrac{a_{by}}{\sqrt{2(1-a_{bz})}} \quad \sqrt{\dfrac{1-a_{bz}}{2}} \quad 0 \quad \dfrac{a_{bx}}{\sqrt{2(1-a_{bz})}} \right]^{T} , a_{bz} < 0 \end{cases} \tag{1}
$$

where ${}^{b}_{i}q$ is the quaternion (a 4-element vector for this application), that rotates the accelerometer from to the global horizontal frame, and $A_{bx}$, $A_{by}$, $A_{bz}$ are normalised accelerometer measurements.

If the authors derive a ${}^{b}q$ for each one of the boulders, then the norm of this quaternion is calculated using the following equation:

$$
n(q) = \sqrt{q_2^2 + q_1^2 + q_3^2 + q_4^2} \tag{2}
$$

where $q_1$, $q_2$, $q_3$, $q_4$ are quaternion elements. I state this calculation here because most of the quaternion operations are different to the typical vector ones. The $n(q)$ of each boulder can be used to check the stability of equation 1 (this is a unit quater- nion, the norm should be approximately equal to 1). The ${}^{b}q$ of each boulder is an

unambiguous metric of orientation change. A time-series of ${}^{b}q$ will show when the boulders have rotated. And the component derivatives of ${}^{b}q$ can give a quantified metric of the speed of this rotation.

As for the linear acceleration, it is enough to just subtract the gravity norm from the raw acceleration norm. It will be much easier to identify major impacts and un- derstand the noise threshold of the accelerometer when it is static. The manuscript states that the sensor at "maximum" settings did not record anything larger than 1g. If that is the total acceleration and there is no compensation for gravity, then the boul- ders did not move (according to the accelerometer), or the data did not transmit at all during transport events for the reason I discuss in my third objection on the analysis of the accelerometer data.

My final suggestion is to try this framework (or at least try to extract global metrics or norms instead of accelerometer axes metics) and then evaluate the use of the moving average filters applied in this work. Most of the IMU errors are not linear, but the reasoning of the filtering methodology used in the manuscript is not unfounded. I think that after the calculation of more global or normalised metrics, many spurious values will be punished or clearly identified in the noise threshold of the sensor which makes them easy to remove.

We agree about the fact that a global metric might in a way be ideal in order to objectively detect a "moved" vs "not moved" boulder. However, at this stage we believe that we cannot necessarily expect the accelerometer graphs associated with moving boulder, to share the same characteristics. This is related to the fact that boulders in different geomorphic contexts will undergo different types of movements (different for timings, velocity, gradual vs abrupt mode, fashion of movement (bouncing, rolling, moving within a mass…), directions and so on). In a way, we could group boulders that shared similar programming characteristics (i.e. average vs max settings) but even within these groups it is reasonable to expect some variability (This is also pointed out by reviewer #2). We have the feeling that obtaining a unique threshold for all boulders in this sense may not be easy, if at all possible, but it is a very interesting topic for future research (particularly for pattern recognition in mature early warning systems), though it is beyond the scope of this paper.

In any case, we made an effort to apply the quaternions method that you suggested to our data and we show example plots in figures 6 and 7 in this document.
Example 1) - In figure 6, we selected a boulder that we could find in the reconnaissance field work after the acquisition period. From our point of view, this boulder was completely stable. What can be observed is that the raw data (top left in figure 6) presents many fluctuations in all axes (though not always simultaneous! i.e. unlikely to be associated with real movement). These fluctuations are within the well-known noise steps (±1 or ±2 steps, or bits, i.e. ±0.184 g or ±0.368 g). However, the data filtered following the moving window three-stages peak reducing method explained in section 3.3, shows three stable axes, which can already be inferred by the raw data graph, when the noise is taken into account. The quaternions were computed and plotted in the graph in figure 6, bottom. What we see here is 1) the noise corresponding to the ±1 or ±2 steps/bits has not been removed, 2) there are large peaks associated to the instances in which the z axis records a value of 0 (axis horizontal with respect to the global inertial frame), due to the structure of the quaternions equation.

Example 2 - In figure 7 in this document we show boulder 4C02, which is also shown in figure 6A of the manuscript. We are confident that this boulder has moved, because there is very strong indication at the beginning of the accelerometer time series (early June 2019) and because this boulder was not found during the field work in October 2019. What we can see again is that the filtering described in section 3.3. of the manuscript removes the well-known peaks between ±1 or ±2 steps of the scale and thus highlights a sharp change, simultaneous in the 3 axes. The quaternions, plotted in the middle panel, do not make this visualisation easier from our point of view and don't immediately appear to offer a better metric to identify movement compatible portions of the time series. This is clear when looking at the bottom panels, that are just a zoom into the first part of the time series, when we think movement occurred. Here, the quaternions do indeed show the

point in which the orientation of the axes is likely to have simultaneously changed as a result of boulder rotation, but the persisting noise, unfiltered by the calculations, has a magnitude very similar to the presumed movement, diluting the interesting part of the signal almost completely.

Unless we misunderstood the application of such method, we do not think that in this case it provides a better way for our identification method and it does not allow to select a threshold above and below which we can say "moved" or "not moved" in a clearer and more objective way.

**Specific comments on the manuscript**

Line 21 ".. and sudden rotations"

Those are difficult to distinguish using the accelerometer measurement only.

The difference here refers to the difference between the slow gradual rotations observed in the landslide embedded boulders and those in the channel for which we only see sharp "jumps" in the time series. We believe that the changes shown in the time series in figure 6 are actually quite clear and occur over a short period of time, thus must correspond to more sudden rotations, as opposed to the minute changes, prolonged in time, seen in the graphs in figure 5.

Line 21, end of paragraph discussing RFID tags

A general note here is that all those techniques work in a "before/ after" event manner. Not suitable for warning.

We disagree with this. The technique as we present it here is not ready yet, it needs further development and we don't claim otherwise, but we believe that this can become indeed very useful for warning purpose, with improved speed of reaction of the accelerometer (decoupled by GPS) and quicker transmission also aided by data compression (this has already been improved in current developments). We are confident that our paper shows that, although we were measuring only at 2 Hz and the accelerometer and GPS were coupled, we were already able to retrieve near-real time (within seconds!) data that indicates a movement has occurred. We have not gone as far as defining a threshold for what would trigger an alarm or not, but our data, with all the capability limitations at the stage of acquisition (and discussed in detail), already show that we could get the data strings corresponding to a boulder orientation change to the server within seconds of it occurring.

Line 91 "Recently, the use of IMUs (Inertial Measurement Unit) has been tested for different applications in 91 the field of geomorphology (e.g. Caviezel et al., 2018 and references therein)."

If the scope here is to refer to previous IMU deployments in geomorphology, I would argue that the first one was from Akeila et al. 2010. And the first milestone from Frank et al., 2014. Those refer to fluvial and coastal transport respectively, and the implementations are quite detailed.

Thank you for these suggestions. Added.

Line 95 "..to reconstruct the path and movement of individual particles in a laboratory flume"

This is not accurate. Unfortunately, it is not possible to reconstruct the path of a stone using a standalone IMU.

We have clarified this point.

Line 109 "...accuracy and precision of the measurement itself, the latter requiring fur- ther development"

Very important comment this one. I sincerely appreciate it.

Thank you. We have been continuing to strive to improve what we can get out of the accelerometer and GPS of our devices and we believe we have already made some

substantial improvements compared to the capabilities we had in April 2019, when we first installed the Nepal network. We are very keen to see future improvements when we will able to install the new devices, which we hope would be in May 2021, but this is sadly still very much uncertain given the global situation.

Line 116 "... the energy thresholds required for remobilisation of different grain sizes"

The term "energy" here is quite misleading. Is this a reference to kinetic energy? I think re-wording this to "forcing" will clarify this sentence

Yes, that's correct. Done.

Lines 132-138

There are paragraphs like this that they need to be summarised more. I know the advantage of a wireless semi-automated warning system is clear. I think that there is a little bit more space on arguing about the benefits of the deployed system that it is required.

Main advantages clarified.

Line 142 "We also demonstrate for the first time the use of this technology in the field of geomorphology, and in a field setup, to monitor the movement of boulders embedded within a landslide and in two debris flow channels."

That is a very strong statement. There are few applications of IMU sensors in fluvial environments. Unless the authors mean the wireless transmission of data through the local network, which is, to my knowledge, a widely applied technology for environmental studies too.

We focus our work on the presence of large boulders in the landscape (this is specified as early as in the title), on the premises that boulders can amplify existing hazards. We think we have given justice to the studies that have used similar technologies both in the field and in experimental set ups. And yes, we do refer to the combination of the use of LoRa technology and smart devices for the detection of boulder movement in a real field setup as opposed to experimental. In this sense and to the best of our knowledge this wireless technology has not yet been applied in the field, in a natural environment (i.e. not under man made conditions, recreated in order to carry out experiments), with the focus being on capturing boulder movement. We are prepared to change this statement if we have missed important literature in this respect.

I think that the sections 2.2 and 2.3 can be summarised. I understand the need to describe the site, but if this paper is more about the deployment of the system, then the focus must be on the method and its validation. This amount of background information just shifts the focus, in my opinion at least.

These sections are 12 and 11 lines respectively.

Line 249 " The sensors are equipped with an accelerometer configured to sample at 2 Hz, as well as a GPS module"

The first question that pops up to mind is " is that enough?" I think we are talking about very gradual motion (before the long intense event). A little bit more discussion about the sampling frequency is necessary.

Just for the sake of clarity, we'd like to specify that we are actually trying to record two different types of movements 1) a long duration, but gradual, with small angular variations for the boulders in the slow-moving landslide 2) potential fast movements of boulders in the channel

during periods of heightened flow.

We believe our data shows a few points quite clearly: 1) 2Hz is enough to indicate the landslide reactivation that reflects on slow boulder motion; 2) 2Hz is not enough if we were to capture rapid boulder movement within an event such as a debris flow. We have commented on this at l. 732 (conclusions section), recognising that this needs improving. In fact, the devices have now been developed to record at a sampling frequency as high as 100 Hz. It is important to bear in mind that what we strive to achieve is a timely identification and alert of hazardous boulder movement *initiation.* Thus, our focus is more on the speed of "wake up" when movement is detected and of the transmission of the relevant strings. Comments added at current l. 726-730.

Line 250 "When movement is detected by the accelerometer, so that tilt or acceleration exceed defined thresholds,.."

This means that the sensor is programmed in a "sleep - wake" routine. What is the frequency of the measurement for the "sleep" state? I assume that the 2Hz sampling frequency refers to the "active" state.

As stated at original l. 303 "The sensors were programmed to send a routine message every 24 hours." In between the 24 hour period, the sensors sleep and do not do anything, unless movement is detected (or spurious acquisitions are triggered by the low angular threshold compared to scale resolution – see above). Clarified current l. 322-324

Line 261 "The depth of the hole allowed for the emplacement of the C-cell batteries and the sensor. After placement, each hole was filled with epoxy resin, sealing the cavity, thus protecting the device from tampering and from the elements..."

Is there any consideration regarding the orientation of the sensor in respect to the frame (the 3D volume) of the boulder?

We mentioned this point in other comments above. Indeed, we should have noted the exact position of the devices within the boulders. Each device was generally placed with the xy plane approximately in a vertical plane (with respect to global inertial frame), thus with the z axis perpendicular to the battery (this is close to horizontal of global inertial frame, but slightly off, depending on the orientation of the drilled hole) and the x axis roughly parallel to the long axis of the battery. In any case, the pitch and roll angles with respect to global horizontal can be seen in the raw data. However, the time available during the field work was heavily affected by many factors. We had to set up the gateway, choose the right position for it, choose the tagging sites, test the connectivity from all sites, solve technical issues with the powering of the gateway and so on. This meant that we had limited time to actual embed and seal with the epoxy the tags and we had to maximise it. Moreover, the epoxy becomes unworkable very quickly and given the limited time available for tagging, the tagging had to happen quickly too. We hope next time to have more time to dedicate to the tagging itself, now that the gateway system is fully functioning and stable. We will then record the exact position of each tag, particularly with respect to the direction. Comments added at current l. 277-283.

Line 277 "The devices deployed in the 2019 season were programmed to not store the data, but to send it immediately, causing the data transmitted during gateway offline time to be lost."

Very common issue, I have been there and it is a very hard lesson to learn. It hurts even if you can repeat the experiment in the next hour.

It really does. In the case of aiming at setting up an alert system, the storage of long-term data is not the one holding highest priority but rather the reaction time and the transmission

speed. However, we hope that our gateway (that has been online constantly since October 2019 – thus only after the acquisition campaign) should suffer less from offline time in future campaigns.

Line 304: "... with an accelerometer event for which activation thresholds can be set for impact forces and for angular variations."

Detail is needed for this threshold

Original l.250 reads: "When movement is detected by the accelerometer, so that tilt or acceleration exceed defined thresholds, collection of GPS and accelerometer data is activated." Further clarified in text current l. 261-265. This can be really anything that is deemed appropriate for the application. Details regarding the thresholds set for this study are already presented in the text: we selected for the low cap scale an impact threshold of 0.4 g and an angular threshold of 5° (original l. 333). The angular threshold was left at 5° for the coarser scale as well (original l. 337).
Whilst this 5° threshold can make sense for the gradual movement that occur in the landslide, it is too low for the coarser scale, as mentioned above. In a way, with our work we also would like to identify the best setting thresholds for different events, this is a really key point also in view of establishing alarm threshold for specific events. This will require tagging many more sites in the future and hope to capture natural events.

Line 309: "In the first case, the values of the three axes are normalised and the measurements essentially represent the static angle of tilt or inclination, thus the pro- jection of the acceleration of gravity, g, on the three axes, ranging between 0 (for a horizontal axis) and ± 1 (for a vertical axis)."

Question a: normalised with the acceleration norm I guess, needs to be clarified. Question b: how the normalised measurement is a direct measure of the tilt angle?

a) Yes. Normalised indicates the fact that the values can only be between -1 and +1 since the projection of g can at most be |1| if a given axis is vertical. Thus, in this type of settings, it would be impossible to capture linear accelerations (i.e. g > 1). Clarified l. 330.

b) This is only the measure of the static tilt. This is indicated by equation 1 of the manuscript. If the value recorded is >0 and <1, a given axis has an inclination with respect to the Earth's reference frame. For a static device, the measured acceleration is simply the acceleration due to gravity. The sine relationship between each axis and its inclination is shown in appendix 3.

Many references are needed here.
This part of the methodology is based on common conventions and on the Euler Angles theory. We don't know exactly what type of references the reviewer deems necessary here. Could this be made explicit?

Line 321 "...and m is accelerometer value recorded on the same axis in g"

Is this a normalised measurement?

This is the difference between two successive readings. The reading is normalized to the max g and can only assume values between -1 and 1, for what we said before, in the

average settings. In the maximum settings it can exceed this value in principle but it never did in our cases for reasons we explain in the discussion. In the snapshots we have of two successive orientations, the accelerometer measures the projection of g.

Line 322 ".. = 0.016 g the corresponding angular variation is of 0.9 if the axis is vertical, but 5.5 if the axis approaches horizontal"

I think the authors use the normalised accelerations and only positive angle 2D changes. This is a very small subcategory of 3D rotations. In addition, it is necessary to describe more equation 1. And the authors need to explain why the resolution changes according the initial orientation of the sensor.

For the purpose of detection of movement trigger, the actual quadrant does not matter (this can be in any case seen by looking at the three axes simultaneously). The focus of this work was to detect potentially hazardous movement as it begins to occur, not to describe the movement accurately. This can be done by timely identifying angular changes even on a single axis.
Once again, the resolution does not change according to the initial orientation of the sensor. Each individual axis simply has a different *sensitivity* to gravity depending on the orientation with respect to the gravity field, for which the axis that lies closer to the vertical of the global inertial frame has higher sensitivity to gravity than axes oriented perpendicularly to gravity do. This has a dedicated appendix (appendix 3). The 2D change in each axis is already an indication of the relative magnitude of rotation each boulder undergoes. The 3D rotation can be calculated using the pitch and roll equations (eq 1 and 2) in this document, but as you point out at the beginning of the review, this is not the aim of our work and we have no information on yaw.
Regarding equation 1, this is a very common relationship used to obtain tilt angle from static accelerometers, in virtue of the fact that, for trigonometry, the projection of the gravity vector on an accelerometer axis produces an acceleration equal to the sine of the angle between the accelerometer axis and a plane orthogonal to the gravity vector (e.g. https://www.biopac.com/wp-content/uploads/app273.pdf, http://aitendo3.sakura.ne.jp/aitendo_data/product_img/sensor/MMA7260Q/MMA7260QT_AN3107.pdf, https://www.digikey.com/en/articles/using-an-accelerometer-for-inclination-sensing). Clarified in text at l. 331-334.

Line 327 "...variability in accelerometer measurements around a stable value, rather than true movement, with this effect becoming more important in sensors programmed with the coarser scale."

The way this is written, it implies that the error and the scaling are programming/ sensor issues. They are not. The differences in scaling appear because the authors used Euler angles (yaw and pitch from what I can understand) which result into singularities. This could be avoided with the use of quaternions.

No, present and talk about the almost raw data, throughout our paper.
We do not imply that the error and the scaling are programming issues, as explained above. Some variability in the accelerometer values we understand is, at least in the case of our devices, inevitable. What we do say is that the extra spurious acquisitions (i.e. not the regular daily, but those triggered by movement) are caused by setting an angular threshold that is lower than the resolution of a particular scale chosen. We do not fully understand what is meant by "The differences in scaling appear because the authors used Euler angles (yaw and pitch from what I can understand) which result into singularities". The spurious peaks are observed in the raw data, well before we calculate pitch and roll angle only for the model boulder visualisation. Moreover, there is *always* a coarsening resolution if we decide to use the maximum 16 g scale, and there would always be

variability in the measurements that are as high as │1│ or │2│ steps on the scale, but the spurious acquisitions are triggered due to the low angular threshold imposed. See, in fact, the quote you inserted in the comment below.

Line 339 "Measurement variability and errors related to the sensors led to spurious data, given the relatively small angular threshold assigned for the highest detectable maximum of 16 g. In other words, given that the step of accelerometer measurement is as high as 0.184 g, a spurious angular variation of more than 5 is often detected even when the boulder is stable, due to intrinsic measurement variability (up to 2 bits)."

This may be an artefact of calculating subsequent orientations and the integration of those. It is very difficult to tell from this presentation.

No, the variability is seen already in the raw data! The graphs presented in figures 5 and 6 show the accelerometer values x,y,z values (with a little smoothing, which is commonly done in time series presentations) referenced to a "zero" only for visualisation. There are no calculations of subsequent orientations. These fluctuations, that are often not even simultaneous in the axes, are simply spurious data associated with acquisitions triggered outside the regular daily cycle, due to the clash between our low angular threshold imposed and the resolution of the measurements. Remember that our devices are activated to send data on movement detection.

Line 346 "within ± 0.184 g of the data point immediately before the window. If any of the values lie outside the ± 0.184 g threshold.."

This averaging corresponds to the "near vertical scenario". What if the sensor is on the "near horizontal" state?

The scale does not change with orientation. What changes is the sensitivity, see appendix 3.

Line 350 "This would mean that a high value would likely be followed by a change in the static angle of tilt of the three axes"

Here, the case of a linear motion without rotation is not captured.

We did not capture linear motions. As we explain in detail in the paper (e.g. discussion section), we think that this is largely related to the time required for the accelerometer to start recording, that at the time of this study was tied to the GPS acquisitions, and to the sampling frequency (2 Hz). We believe we never recorded the "full" movement, thus nor values above 1 g. We only record a "before" and "after" orientation, but within seconds of the change happening (e.g. B# 4C02). We think that if we observe a sudden change in the orientation of the three axes, it is safe to assume that a movement of the boulder must have occurred.

Line 351 "Therefore, it is unrealistic to have a peak value followed by a value equal to that observed before the peak."

If the sampling frequency is at 2Hz, this is potentially true for the larger boulders.
But not safe to assume for the smaller ones.

By simply holding an accelerometer in the hand, it is possible to see that one can impart an

acceleration in one axis only, e.g. z axis when this is parallel to gravity, by lifting or dropping quickly the device. It is extremely unlikely that the movement of a boulder (for as much as it may be a small boulder – and the smallest boulder we tagged has a volume of 0.018 m3 and thus an estimated weight of around 50 kg) in a debris flow could be caused by an acceleration and deceleration in one direction that wouldn't involve the other axes. Even more so because we sample at 2Hz in this study: we think it is very unrealistic that the 0.184 g peaks we observe are the result of a "small push" in one direction and then a sudden stop. The boulder would have to accelerate in one direction and stop without changing its orientation… The boulders are in a natural environment, with high roughness and potential interactions between each other.

Line 359 "The accelerometer readout in the current version of the software is tied to a GPS acquisition, this means that although the accelerometer is measuring as soon as movement is detected, the acquisition is obtained only when the GPS has successfully retrieved the position."

This is not an issue, as long as there is a clear description of when the movement is detected (time and acceleration threshold)

This is actually a very big issue, as explained in detail in our discussion and as pointed out by reviewer #2 as well. If a boulder is picked up by a flow and moves downstream a few meters within a few seconds, we will not see this if the GPS takes 120 seconds or more to acquire a position (or fail) before allowing the accelerometer to begin its recording. We think that this is one of the two most important reasons why, the three boulders we show moved in the channel, have not captured the full movement: the acquisition was held up by the GPS. This problem has now been solved with the most recent version of the devices.

A quantification of how "rough" is this estimate here would be useful.

Added l. 432.

Line 386 This data is used in a qualitative way for comparison with and validation of the accelerometer data obtained with the wireless devices and, despite the qual- itative approach, this data provided a quite detailed overview of the days in which movement occurred.

It is important to stay here how the authors associate the geomorphic change with the accelerometer data. It is necessary to recognise that the comparison refers to two vastly different time resolutions.

This is obvious. However, we hope you would agree that scan data and photos clearly show that movement (order of a few m) has occurred in the area, and that therefore it is reasonable to assume that some of our boulders were subjected to such movements. In a way, the fact that we know some geomorphic processes were ongoing both in the channel and in the landslide, shows that it is far less likely that the data of our accelerometers, that we argue is compatible with boulder movement, is a mere coincidence. This is particularly true for the boulders in the landslide that show movement precisely during the periods in which the photos (see timelapse in the video in the supplement) show important sliding phases. Added comment l. 641-646.

Line 421: "The values of each axis are recalculated to show the deviation from the original position for visualisation purposes, rather than the actual values measured (hence all raw data curves begin at 0, and the smoothed curves around zero, due to the smoothing)."

This can be very misleading. The initial orientation is crucial for interpreting the accelerometer data.

We agree that the initial orientation can be an important piece of information, but this completely misses our main point. We are trying to identify the *onset* of potentially hazardous movement. We do have the raw, original x, y, z data but we do not attempt to produce a realistic reconstruction of boulder rotation, as this was not part of our original objectives. We greatly prefer this visualisation, with all axes starting at a "zero" position, to show only *the point at which* movement is likely to occur and to be seen in the data. The three axes plotted on their original values would simply make the plots very hard to read because the scale would go from -1000 and +1000 and the internal variations would look very small in a normal size figure. The content of the graphs is the raw data, with a bit of smoothing, referred to a common zero, therefore the information it contains should not be misleading.

Line 434 "Fig. 4 and 5 show that the movements of boulders within the landslide not only differ in the magnitude of the angular variations recorded, which is an order of magnitude higher for B A226 and B 9A41 in comparison to other boulders, but also in the evolution with time."

There needs to be an objective, quantified metric for this comparison. There is plenty of data.

We are looking at the _actual_ data, therefore observing the difference between a change throughout the observation period of 20 mg or a change of 200 mg can allow for a reasonable, quantitative comparison.

Line 461 "These boulders were programmed to retrieve actual g values (as opposed to normalised values) and forces up to 16 g."

This needs be highlighted and clarified much earlier in the manuscript

This comment is rather surprising because we have a whole section earlier in the paper (section 3.3) to explain all the settings used in the study. In this section we do say that boulders that we thought might undergo rapid movement were programmed with the scale capped at 16 g, because we hoped this might have allowed us to retrieve large impact forces (if the GPS didn't exert such a drag on the system).

Line 473 "We do not observe forces > 1 g for any of the sensors programmed with the maximum settings, despite the ability of the sensors to detect up to 16 g. This is consistent with a lack of debris flow activity recorded by cameras or seismome- ters, the more prolonged activity of which would have generated sustained boulder movement, beyond the time needed for GPS acquisition as explained below"

This is not compatible with the detection of linear accelerations. Compensation for gravity is required.

As mentioned before, we do not record linear acceleration. We did not have the chance to record movement as it unfolded due to a combination of technical capability and environmental conditions. See response to earlier comments and original l. 632-633. If we state in the quote in your comment just above that we do not observe values > 1 g, and we explain that this is because we don't have prolonged activity that would spill beyond the GPS acquisition period, why should there be the need to compensate for gravity when we believe that what we observe is always the static measurements (even if in some cases most likely immediately after movement occurred, and this we know

because it is within seconds of a previous acquisitions and acquisitions in which observe change in orientations were triggered) ?

Lines 458-459 require further quantification

We are not sure we understand this comment. Is this referred to the "important sliding" with references to the landslide movement visible in the image? Figure 7A and 8A shows arrows that represent the movement magnitude in the plane of the image. We already mentioned in the original version 2 m of displacements observed in the images in the mid to lower parts of the landslide and 1 m of displacement at the headscarp seen in the scan data.

Lines 508-515

I know that those deployments are extremely difficult and they don't always go to plan. But the discussion of the GPS data here is not very useful. It is both too long and not directly feeding to the interpretation of the data. My honest opinion is that the paper would benefit if the GPS data were not discussed in the main body. It is probably useful for an appendix to demonstrate the deployment, but there is no clear quantitative information that can be extracted from here.

We appreciate your comment, but although we are aware that you believe the length of the GPS acquisition was not an issue, we on the contrary argue that was one of the main factors that held up quicker acquisitions of movement data. This is the main reason why, we believe, we were not able to detect linear accelerations. Perhaps this point should, in fact, emphasized further, so that the reader would not get the impression that we need to compensate for gravity linear accelerations that we did not record.

Line 543 "The movements observed for the boulders scattered on the landslide body and embedded within the material can be described as small angular variations that occurred gradually during the season"

Those are the type of statements that require further quantification.

The graph in figure 5 show when the changes in x,y,z occurred in the season, for how long they continued and what magnitude ($g^{-3}$) they were. These movements are already described in the results section 4.1.

Line 550 "...show higher magnitude of the angular variations with respect to other boulders (Fig. 5F-G)."

How much higher?

This is shown in the raw data in figure 5. The graphs clearly indicate that the uppermost boulder (F3CE) sees variations that range between │10│ and │30│ mg, whereas boulder 9A41, further down the slope, between │30│ and │200│ mg.

Line 552 "..follows a spatial and temporal pattern"

This is a very strong statement. I would definitively require some statistical justification.

With 6 boulders moving with the landslide, it would be extremely difficult to produce any meaningful statistics. We do however note, record and point out the fact (perhaps a mere coincidence?) that there appear to be some sort of pattern visible in the map in figure

4. This is true only with reference to the landslide, *not* the channel, and we explain why we think this is not a mere coincidence and why this might be the case in section 4.1. This highlights, in our opinion, the potential to extract information on landslide characteristics and mechanics if a denser network of nodes were to be deployed.

[Figure]

Line 588 "What controls this behaviour is not the fact that the sensors were pro- grammed to detect the maximum force or the static tilt respectively, but rather the scale that was chosen and associated with the two settings types."

That can be true, but it is not the only reason for increased noise. My first guess would be de-callibration or humidity. Those sensors are very temperamental. And most of the noise is traditional, random AC-DC circuit noise.

We appreciate your comment, however, we do think that the settings imposed on the boulders cause the differences observed. A scale that has a resolution of 0.184g is bound to have a higher level of noise than a scale with resolution 0.016 g. We do not observe increased noise *in time* for a given sensor. Moreover, although we fully agree that temperature for example might play a role in affecting the devices, we think that if this were to be a determinant factor, we would observe a high variability between sensors (e,g, shadows and localized effects) whereas humidity is unlikely to play a role on a device fully embedded in epoxy resin. Our data does not show a high variability between devices. See figures 1-5 in this response. You can see that all the boulders programmed in the same way (i.e. max settings) and that did not appear to move, show similar noise patterns. In truth, there is some variability (some acquired more fixes than others, some batteries discharged earlier than others), but the noise level is indeed very similar and comparable between devices and through time.

Line 602 "Future improvements of the accelerometer accuracy, resulting for example from the activation of the 9-axes IMU present in the hardware of the devices, could reduce this problem"

This is very true. Especially the gyroscope measurement will be very useful for this type of measurement.

Indeed. This work is planned in the future, including activation of gyroscope and magnetometer, though it requires a lot of effort on the software development side.

Line 616 " The high positional errors and the important battery expenditure make the current GPS module not fit for the purpose of tracking boulders in rugged terrains."

That is also very true. We need to investigate or come up with alternative tracking techniques for remote areas.

Yes, and if we were to use boulders to get alerts on important geomorphic events that involve boulder movement, a full tracking with this technology is not even necessarily what we need to focus on. We think that for the early warning side of things, we have to make sure that we can quickly activate the devices and transmit the relevant data. For each case there would need to be serious thought put into the selection of sites (how far upstream of key sites to allow for some viable alert time).   Moreover, we would need to understand 1) what are appropriate thresholds for hazardous movements 2) what are patterns in the accelerometer time series that correspond to particular events.

Line 630 " This may explain why, although the boulders in the channel were pro- grammed to detect high forces, they never show accelerometer values higher than 1 g (either negative or positive)."

This is true, but I think there is also an artefact of the processing followed here. I am not sure the authors will pick up very high inertial force. The boulders are quite large and heavy. Maybe 1g is too small though (see notes above).

We are not sure we fully understand. Boulders are large and heavy, however, anything that would be able to displace them, would also require to produce a high impact force. We decided to use the 16 g scale, because this was our first attempt and we decided to allow ourselves to capture really large forces (in case of large events, which did not occur). In reality, we could have probably used a much lower cap, such as 4 or 8 g.

The last section of the discussion (5.1) is useful but I think it could be summarised a lot. A table of prons and cons would be a good addition.

Line 730 "...but that in the future are expected to replace the need for an accurate GPS."

I would strongly oppose that. There is no evidence that the available IMUs will be suitable for standalone tracking anytime soon. Unless, the authors refer to mili- tary grade optical sensors which cost £10k each. If that is the case it is necessary to provide some specs.

We rephrased this. The purpose of our work, as stated above, is not that of tracking long term the position of a boulder, but rather to timely identify when and why a boulder begins to move (and, potentially, what are the dynamics of fast but periodic movements, with higher sampling frequency now available).

GM

[Figure]

Fig. 1. Analysis of stability of boulders 2F7F, B0C6, 6EAC. Panels on the left show the accelerometer data for x, y, z axes. Histograms show the frequency of measurements that are below or above the initial value. The five sequences represent equal sample size of accelerometer reading, with time increasing from sequence 1 to sequence 5.

[Figure]

Fig. 2. Analysis of stability of boulders 8AA1, 017E, 036E. Panels on the left show the accelerometer data for x, y, z axes. Histograms show the frequency of measurements that are below or above the initial value. The five sequences represent equal sample size of accelerometer reading, with time increasing from sequence 1 to sequence 5.

[Figure]

Fig. 3. Analysis of stability of boulders 625C, 4916, 9773. Panels on the left show the accelerometer data for x, y, z axes. Histograms show the frequency of measurements that are below or above the initial value. The five sequences represent equal sample size of accelerometer reading, with time increasing from sequence 1 to sequence 5.

[Figure]

Fig. 4. Analysis of stability of boulders 3BDE, B8E8, B45D. Panels on the left show the accelerometer data for x, y, z axes. Histograms show the frequency of measurements that are below or above the initial value. The five sequences represent equal sample size of accelerometer reading, with time increasing from sequence 1 to sequence 5.

[Figure]

Fig. 5. Analysis of stability of boulders ED43, D7A0. Panels on the left show the accelerometer data for x, y, z axes. Histograms show the frequency of measurements that are below or above the initial value. The five sequences represent equal sample size of accelerometer reading, with time increasing from sequence 1 to sequence 5.

[Figure]

Fig. 6. Example, boulder B0C6. This boulder did not move (stable). Comparison between raw data (top left), filtered data (top right), calculation of quaternions and quaternions norm (bottom).

[Figure]

Fig. 7. Example, boulder 4C02. We think this boulder moved (indicated by data and not found during field work, see Fig. 8C in manuscript). Comparison between raw data (top left), filtered data (top right), calculations of quaternions and quaternions norm (middle), bottom panels, zoom into first 200 acquisitions (including what we think is indication of movement), with raw (left) and quaternions (right). This boulder is shown in figure 6A of the manuscript.

Earth Surf. Dynam. Discuss.,
https://doi.org/10.5194/esurf-
2020-78-RC3, 2020

[Figure]
The presented work focuses on the deployment of accelerometers and its real-time data transmission as possible low cost means of surveillance for large single blocks to identifiy mass movement associated with landslide type of rock slides. It presents a substantial an thoroughly carried out field measurement campaign and careful data analysis. The use of IMU technology for boulder tracking and its possible applications for early warning systems is a highly relevant topic. The comprehensive presentation of this pilot study definitevly merits publication after some minor revisions.

Generally, the presentation of the entire work is very nicely done. I also have to thank the authors to present a carefully edited and proofread manuscript, which made reading easy and enjoyable.

Dear reviewer #2, thank you for taking the time to read our manuscript and for judging our work worthy of publication. We also thank you for offering interesting cues to references and use of techniques that we had not previously included. We tried to address the points you have raised in the manuscript and we give our response to each comment in teal below.

In the following some content and technical suggestions for improvement and addi- tional context are provided. IMHO the manuscript would benefit of some remarks on remaining challenges and disadvantages of IMU tracking/signal processing.

Introduction l41: Large boulder movement rarely comes isolated. While the approach to use large boulders as particle marker for mass movements with modern technology is new, the general statement that the motion of large boulders and its damage potential is not discussed in literature may be a bit exaggerated.

We did not mean to imply that hazards that involve the movement of large fragments have not been studied. The presence of boulders of given sizes in given proportion has not been, to the best of our knowledge, directly accounted for in hazard assessments of landslides and floods. However, we deleted this sentence since it is more relevant for another part of our work that is not included in this publication.

L54: large boulders can be detected via RADAR/LiDAR technology, which is truly re- mote. The target boulders here predominantly are early warning signs

L64: State-of-the art RADAR (no interferometric RADAR of course) techniques are able to deliver real-time data for immediate mitigation actions such as road closures etc. See https://ui.adsabs.harvard.edu/abs/2020EGUGA..22.5138W/abstract for the lack of better reference sake.

Thank you for this, mentioned.

General remark: With all the advantages listed for the IMU technology applied, one crucial disadvantage needs to me mentioned: The installation of the sensors do require physical presence at the block. While this may not be a problem for large boulder instrumentation in slowly evolving mass movements, this is certainly a major drawback to deploy the presented technique in active sites.

It is certainly a drawback (mentioned in text at l. 750) and it is true that it might be impossible to tag particularly dangerous sites. However, we did tag active sites (not rockfall sites) and also this technology could be used to tag upper reaches of catchments (e.g. km upstream of sites affected by flash floods). Equally, to install a monitoring network that requires the use of ground based LiDAR or RADAR, a base station has to be placed with line of sight of an active site. This is also not feasible in many instances. The answer is probably that no technique is perfect for all cases, but each case would have to be evaluated carefully to decide what technique is more suitable (also in terms of economic efforts). The network type we propose has the enormous advantages of becoming cheaper in the future and to allow for activation on movement.

Methodology 3.1 Network setup and components Really nicely presented methodol- ogy!

Thank you for this nice acknowledgment.

Notation remarks:  ć Generally throughout the manuscript, change the notation of the local gravitational field of Earth to texit{g} or $g$ as it denotes a physical constant usually denoted in italic font. This also removes the ambiguity of mg and mg.  ć The same holds for x,y and z axis, variables denoted by italic characters. Any given coordinate system is given by its n-space.

We have changed the notation of g and x,y,z to italics, we hope we have correctly interpreted this comment.

3.2 Choice of tracked boulders l298 coherently collectively/mutually. Coherence would imply that the motion pattern is the same, as a laser has coherent wavelengths. Large boulders can move with the landslide but usually succumb to a slightly different kinematical regime. True coherence in nature is extremely rare.

We mean "as a whole". Clarified throughout.

3.3 Sensor Settings l323ff replace the "~" with approx or the word roughly, about, etc.. Tilde means "similar to" and is usually used in plain mathematical context.
Done.

l352 maybe add "before the peak when sampled at 2 Hz." If sampled at higher fre- quency, such double or three peak hits are not that uncommon.

True. Peaks would not be uncommon if movement occurred in which case one could expect to observe a sequence of simultaneous peaks in all axes and with different values attained during or after the end of the movement sequence (see a comment above in response to reviewer #1). But surely the fact that we are sampling here at 2 Hz, as you say, makes the peaks we see even less likely to be associated to real movement. Suggestion added.

4 Result Thorough presentation of the results. Only notation of axis and g and "~" characters would need some attention.
Done.

5 Discussion Validation of motion is partly done via camera imagery. While I would agree that only tilting motion of an embedded rock is not feasible to be detected via imagery, I would argue with the progress in resolution an image processing, a pixel tracking via cross correlation analysis of interval imagery might well track slow motion onsets. The spatial resolution is then given by the camera's resolution. Just one of a zoo of cross-correlation papers (https://nhess.copernicus.org/articles/17/2143/2017/ )

The detectable grain size would highly depend on resolution/distance and the detectable movement also on the movement magnitude. Here we are talking of boulders imaged at a distance of approximately 600-700 m (depending on exact point within the network). This, according to the resolution of the camera, should give a pixel size for the scene acquired of about 15 cm. Indeed, we can see quite clearly large boulders in the channel. The landslide area however, as it can be seen both in figure 7A and in the video provided as supplement, is at a relatively low angle with the LOS of the camera. The camera looks towards ESE (approx. 119°) and the direction of the plane of the landslide is NW (approx. 327°). Finally, the tilting of the boulders in that region is shown in the accelerometer data to be of a few degrees only. Slow motion onset of the whole landslide mass is indeed well visible and this

could surely be tracked with appropriate pixel offset techniques. Though this would be useful, it is beyond the scope of our paper.

L668 while in the introduction the https://nhess.copernicus.org/articles/17/2143/2017/ heritage of animal tracking is mentioned, a com- parison with state of the art logistic tracking devices such as MSR sensors or trusted global devices (just to name two), would be interesting. Modern logistic shock tracker do also work with acceleration and angular velocity IMUs and sometimes even come with satellite network coverage to send the reports.

L688 As stated by the authors, independce of GPS/GNSS signals is of paramount importance.

Yes, and we have now achieved this with the new development. Thanks for acknowledging this.

L731 Accurate position information from IMU sensor integration requires sophisticated post-processing procedures in order to minimize integration error accumulation. This is feasible in case of periodic motion or motion patterns, where at specific positions in time a zeroing of the errors is possible.

If this is not the case, accurate position tracking via IMU is extremely challenging, especially for fast motion. If GNSS (maybe refer to GNSS than GPS alone, as there are many other systems in the sky then GPS only) measurements will become obsolete in the future, one will see.

Rephrased.

---

## Editor Decision (ED1)

Dear Authors,

I have now examined the reviewers' comments, your replies, and the revised manuscript. Both reviewers stress the importance of this work, presenting the use of IMU technology for tracking boulders and thus landslide movements, in addition to recognizing the substantial effort that went into this study. I think you did a very nice job addressing the comprehensive comments. However, there are a few minor comments I would like you to address before the manuscript is ready for publication.

- You clarify in the response to reviewer 1's comment that "the main point of this paper is not that of offering a 3D representation of boulder movement" and that "the main point of this first paper is to show that this technology might mature quickly in the near future to provide real time data on the initiation of hazardous boulder movement, which we believe our data already highlights, despite the capability limitation at the time of data acquisition." However, in the Discussion it could be interpreted as if you are contending that the main point of your paper is to show 3D representations of boulder movement ("this indicating the potential of the technology used for detecting both gradual angular variations and changes in boulder orientation associated with rapid movements in real or near real time"). Please clarify your text in order to not overstate the realistic objectives/outcomes of this paper.
- L114: This sentence is missing a word. Either add 'us'/'researchers' after 'allow' or change 'to investigate' to 'investigation of'
- L140: This sentence is missing a word. One suggestion is to add 'providing' before 'the potential' (remove comma before 'and')
- L169: 'PGA' has not been defined previously in the text.
- Figure 1: Pink box showing 'Zoom Fig. 4' is quite hard to see.
- L287: change to 'a 4-panel solar system'
- L304: Change to either: 'have a b-axis of' or 'have b-axes of'
- L324: A word is missing here: 'in the following _______'. Should this be text, paragraph, section?
- Figure 3: Are the measurements given on the scale bars the b-axis measurements? If so, please add that information in the caption. If not, why choose such different scale bars and not use a consistent measurement of 0.5 or 1 m?
- Figure 5: Please describe the different elements of the figure in alphabetical order (i.e., A & B before C-G)
- Figure 6: As the reviewer required additional explanations of how the representations were created in Fig. 6B, D, F, I think it could benefit readers to include equations (1) and (2) (from the 'Author response') together with a brief explanation in the text.
- Figure A3. The yellow text and arrows are quite difficult to read; please choose a more distinct color.

Best Regards,

Lina Polvi Sjöberg

---

## Author Response (AR2)

Dear Authors,

I have now examined the reviewers' comments, your replies, and the revised manuscript. Both reviewers stress the importance of this work, presenting the use of IMU technology for tracking boulders and thus landslide movements, in addition to recognizing the substantial effort that went into this study. I think you did a very nice job addressing the comprehensive comments. However, there are a few minor comments I would like you to address before the manuscript is ready for publication.

Dear Dr Polvi Sjöberg,

thank you very much for taking the time to examine the revisions and our response to the reviewers and for giving us some additional suggestions. We have addressed your comments and hopefully improved the manuscript to the required standard.

- You clarify in the response to reviewer 1's comment that "the main point of this paper is not that of offering a 3D representation of boulder movement" and that "the main point of this first paper is to show that this technology might mature quickly in the near future to provide real time data on the initiation of hazardous boulder movement, which we believe our data already highlights, despite the capability limitation at the time of data acquisition." However, in the Discussion it could be interpreted as if you are contending that the main point of your paper is to show 3D representations of boulder movement ("this indicating the potential of the technology used for detecting both gradual angular variations and changes in boulder orientation associated with rapid movements in real or near real time"). Please clarify your text in order to not overstate the realistic objectives/outcomes of this paper.

We tried to made this a little clearer in the first paragraph of the discussion. What we mean is however what is already written in this paragraph. Basically, even though we do not aim here at offering a 3D representation of boulder movement as it unfolds, we are able to capture the moment in which the movement initiates. Movement onset occurs in the form of some change in the x,y,z orientation of the accelerometer axis. This change, as we show in the results, can be small and gradual or much larger (and we assume the latter corresponds to larger movement, the full development of which we don't capture due to GPS delay and sampling frequency).

- L114: This sentence is missing a word. Either add 'us'/'researchers' after 'allow' or change 'to investigate' to 'investigation of' Done.

- L140: This sentence is missing a word. One suggestion is to add 'providing' before 'the potential' (remove comma before 'and') Done.

- L169: 'PGA' has not been defined previously in the text. True, made explicit.

- Figure 1: Pink box showing 'Zoom Fig. 4' is quite hard to see. This has been made lighter and thicker.

- L287: change to 'a 4-panel solar system' Done.

- L304: Change to either: 'have a b-axis of' or 'have b-axes of' Done.

- L324: A word is missing here: 'in the following __________'. Should this be text, paragraph, section? Done.

- Figure 3: Are the measurements given on the scale bars the b-axis measurements? If so, please add that information in the caption. If not, why choose such different scale bars and not use a consistent measurement of 0.5 or 1 m? The 1 m scale will be different for each photo anyway, which will make the figure looks less tidy. We decided to show the dimension of the boulder in the plane of the photo for a more immediate reference. We have changed the figure using a consistent 1m reference, but we would prefer the original version.

[Figure]

- Figure 5: Please describe the different elements of the figure in alphabetical order (i.e., A & B before C-G) Done.

- Figure 6: As the reviewer required additional explanations of how the representations were created in Fig. 6B, D, F, I think it could benefit readers to include equations (1) and (2) (from the 'Author response') together with a brief explanation in the text.

This is fine, perhaps it would be best to have it as a small supplement (appendix 4), since it is only the application of the equations of pitch and roll angles to our accelerometer data.
Also, if you think it may be even better, we considered the possibility to remove figures 6B, D, F if these add more confusion than they help visualise what could have happened to the boulders.

- Figure A3. The yellow text and arrows are quite difficult to read; please choose a more distinct color. Changed.

On behalf of the authors,
Benedetta Dini